# Focus-Then-Reuse: Fast Adaptation in Visual Perturbation Environments

**Jiahui Wang**[1,2]*, **Chao Chen**[1,2]*, **Jiacheng Xu**[3], **Zongzhang Zhang**[1,2]†, **Yang Yu**[1,2]

[1]National Key Laboratory for Novel Software Technology, Nanjing University, Nanjing, China
[2]School of Artificial Intelligence, Nanjing University, Nanjing, China
[3]Nanyang Technological University, Singapore
{wangjh,chenc}@lamda.nju.edu.cn, jiacheng005@e.ntu.edu.sg,
{zzzhang,yuy}@nju.edu.cn

## Abstract

Visual reinforcement learning has shown promise in various real-world applications. However, deploying policies in complex real-world environments with visual perturbations remains a significant challenge. We notice that humans tend to filter information at the object level prior to decision-making, facilitating efficient skill transfer across different contexts. Inspired by this, we introduce Focus-Then-Reuse (FTR), a method utilizing a novel object selection mechanism to **focus** on task-relevant objects, and directly **reuse** the simulation-trained policy on them. The training of the object selection mechanism integrates prior knowledge from a vision-language model and feedback from the environment. Experimental results on challenging tasks based on DeepMind Control Suite and Franka Emika Robotics demonstrate that FTR enables rapid adaptation in visual perturbation environments and achieves state-of-the-art performance. The source code is available at `https://github.com/LAMDA-RL/FTR`.

## 1 Introduction

Visual Reinforcement Learning (RL) has achieved breakthroughs in a wide range of real-world applications, including robotic manipulation [1–3] and autonomous navigation [4, 5]. However, bridging the gap between simulation and real-world environments remains a pivotal challenge. In this work, we focus on distracting background, which is a typical form of visual disturbance. Taking robotic grasping as an example, although existing methods can achieve good performance in simulation [6, 7], directly deploying the learned policies in the real world may suffer from performance degradation when there exists complex backgrounds [8, 9].

Recent studies have explored ways to address the challenge of deploying policies in complex environments with visual perturbations. Existing approaches can be categorized into three groups: (1) training policies directly in noisy real-world environments [10–12]; (2) learning generalizable policies that are robust to environment variations [13–22]; and (3) adapting pre-trained policies to target environments with visual perturbations [23–25]. Each of these approaches presents potential limitations. First, policies trained directly in visually disturbed environments often incur high costs and risks, and the policy may generalize poorly to unseen settings. Second, generalization-based methods rely on techniques such as data augmentation to simulate real-world variations during training [26]. However, if the training process fails to capture the diversity of the deployment environment, severe performance degradation may occur. Lastly, existing adaptation methods struggle to

---

*These authors contributed equally.
†Zongzhang Zhang is the corresponding author.

fully preserve policy performance achieved in simulation. We concentrate on an adaptation approach that aims to preserve the source policy performance with minimal degradation.

Inspired by the stark contrast with the limitations of existing adaptation methods, we investigate human skill transfer. Notably, humans possess a natural ability to transfer skills across diverse contexts. We believe this ability can be attributed to two key aspects. First, during perception, humans perform object-level filtering to distinguish between task-relevant and irrelevant objects, effectively reducing the complexity of information processing [27]. Second, in decision-making, humans can leverage both prior knowledge and environmental feedback to guide their actions. Initially, object selection is driven by prior knowledge; if the outcome is unsatisfactory, humans will iteratively adjust their selection policy based on environmental feedback until the task is completed [28].

Humans acquire prior knowledge by summarizing a large number of past experiences. Similarly, foundation models gain strong prior knowledge through pre-training on large-scale and diverse datasets, achieving remarkable success in natural language processing [29, 30] and computer vision [31–33]. Recently, many studies have also explored using foundation models as a source of prior knowledge for downstream tasks [34, 35].

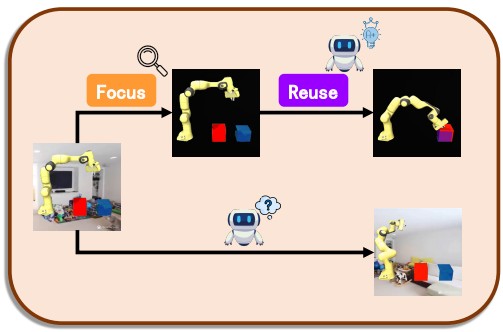

Figure 1: Directly deploying the learned policy in real-world environments may degrade performance. We propose Focus-Then-Reuse, a method that utilizes a novel object selection mechanism to focus on task-relevant objects, and directly reuses the learned policy on them.

Inspired by humans' ability to transfer skills and the success of foundation models, we introduce a new paradigm for visual domain adaptation RL. We propose Focus-Then-Reuse (FTR), a method that directly applies a simulation-trained policy to observations focused by a novel object selection mechanism (Fig. 1). The object selection mechanism consists of a trainable segment selector, a fixed segmentation model, and a fixed tracking model. The training of the segment selector synthesizes prior knowledge from a Vision-Language Model (VLM) and feedback from the environment. Experiments on DeepMind Control Suite and Franka Emika Robotics show that FTR facilitates rapid policy adaptation from clean training environments to visually perturbed target environments and achieves state-of-the-art (SOTA) performance.

We summarize the contribution of this paper as:

- We introduce a novel Focus-Then-Reuse framework for policy adaptation, with the focus stage filtering task-relevant objects and the reuse stage employing a fixed policy for faster and more stable adaptation.
- We propose an adaptation RL method that integrates supervised learning and reinforcement learning to synthesize VLM's prior knowledge and environmental feedback.
- Experiment on challenging tasks demonstrates that our method facilitates quick and effective policy adaptation to visual perturbation environments, achieving SOTA performance.

## 2 Related works

The pursuit of robust policy deployment across visually perturbed domains has catalyzed research interest in recent years. Some works try to train policies in visually distracted environments directly. DBC [36] first explores this pathway by learning a representation robust to distractions. Building on DBC, $Q^2$-learning [11] decouples policy learning from behavioral metric learning for stable training. Some model-based methods [12, 37–39] explicitly recognize task-relevant parts. However, policies trained directly in visually perturbed environments often suffer from high training costs and risks, as well as limited generalization to other scenarios.

Generalization methods explore ways to enhance policy performance, which can be classified into three types: data augmentation, inductive bias, and learning invariances [26]. Data augmentation methods [14–18] apply techniques like cropping and color jittering to training images to reduce

the distribution gap between training and testing. Inductive bias methods [19–22, 26] incorporate assumptions about task-relevance (e.g., foreground is of higher importance), among which Sim-GRL [13] achieves the SOTA performance. Invariance-based methods [40–45] focus on extracting information consistent across diverse training environments. We point out that generalization methods face two issues. First, added perturbations or inductive biases may hurt training performance. Second, when facing unseen disturbances, the performance may drop significantly.

Domain adaptation RL focuses on transferring policies trained in the source domain to the target domain [46–50]. Of particular relevance to this paper is visual domain adaptation RL, which remains relatively under-explored. PAD [23] is the work most closely related to our setting. It introduces a self-supervised objective during training and performs online adaptation at deployment through this objective, enabling policy adaptation to target domains with simple backgrounds. Other studies have explored policy adaptation to target domains with varying camera viewpoints [24], changes in color and object scale [25] or new visual dynamics [51]. However, we suggest that current adaptation methods fail to fully recover the policies' performance, with severe degradation in challenging environments like video background [17].

Foundation models can leverage their prior knowledge to support perception and decision-making in downstream tasks [52, 53]. For perception, some approaches [54, 55] employ foundation models as pre-trained feature extractors, while others [10, 56–59] incorporate promptable segmentation models to enhance scene understanding and representation learning for visual RL agents. For decision-making, a common approach is to employ foundation models as reward generators to provide learning signals for policy optimization [60, 61]. Our method leverages foundation models to assist RL from both aspects. On one hand, we use pre-trained segmentation and tracking models to process complex inputs in a zero-shot manner. On the other hand, we employ a Vision-Language Model (VLM) as a supervision signal in policy training.

## 3 Preliminaries

**Reinforcement learning**  Traditional RL considers the task in the form of a Markov Decision Process (MDP) $M = (\mathcal{S}, \mathcal{A}, \mathcal{P}, R)$. $\mathcal{S}$ is state space, $\mathcal{A}$ is action space. $\mathcal{P} : \mathcal{S} \times \mathcal{A} \times \mathcal{S} \to [0, 1]$ is the transition function that defines the conditional probability distribution $\mathcal{P}(s_{t+1}|s_t, a_t)$ over next states given state $s_t \in \mathcal{S}$ and action $a_t \in \mathcal{A}$ at time $t$, $R : \mathcal{S} \times \mathcal{A} \to \mathbb{R}$ is a reward function. RL aims to learn a policy $\pi(a|s)$ that maximizes the expected discounted cumulative reward $\mathbb{E}_\pi \left[ \sum_t \gamma^t r_t \right]$ where $\gamma \in [0, 1]$ is the discount factor. In visual RL tasks, the agent can only have access to the observation $o_t$ rendered from $s_t$, and thus the policy becomes $\pi(a|o)$. In this paper, we use Proximal Policy Optimization (PPO) [62]. PPO is an effective algorithm for solving general RL problems. Given policy $\pi_\theta(a|s)$ and value function $V_\psi(s)$, PPO maximizes the following objectives:

$$\mathcal{L}(\theta) = \mathop{\mathbb{E}}_{(s,a,r,s') \sim \pi_{\theta_{old}}} \left[ \min \left( r(\theta) A^{\pi_{\theta_{old}}}(s, a), \text{clip}(r(\theta), 1 - \epsilon, 1 + \epsilon) A^{\pi_{\theta_{old}}}(s, a) \right) \right],$$
$$\mathcal{L}(\psi) = \mathop{\mathbb{E}}_{(s,a,r,s') \sim \pi_{\theta_{old}}} \left[ V_\psi(s) - (V_{\psi^-}(s) + A^{\pi_{\theta_{old}}}(s, a)) \right]. \tag{1}$$

Here $r(\theta) = \frac{\pi_\theta(a|s)}{\pi_{\theta_{old}}(a|s)}$ is the probability ratio measuring how much the current policy deviates from the old policy used for data sampling, while the clip($\cdot$) operation prevents overly large policy updates. The hyperparameter $\epsilon$ defines the clipping boundary. The value function $V_\psi(s)$ is a parameterized estimate of the expected discounted cumulative reward starting from state $s$ and following policy $\pi_\theta$ thereafter. The advantage $A^{\pi_{\theta_{old}}}(s, a)$ is usually estimated from $V_\psi(s)$ via Generalized Advantage Estimation (GAE) [63]. $\psi^-$ denotes the target network parameter, which is periodically synchronized with $\psi$ for stable training.

**Visual foundation models**  Visual foundation models are commonly pre-trained on massive datasets and can adapt to various downstream tasks in a zero-shot way. In this work, we focus specifically on foundation models designed for image understanding [33], image segmentation [64], and object tracking [65]. For image understanding task, we use Qwen-VL-Max [33], a multimodal version of the Qwen large model series that outperforms current SOTA generalist models on multiple vision-language tasks. It takes a combination of image and text as input and generate structured textual outputs following a specified format (e.g., JSON). For image segmentation and object tracking, we use SAM 2 [32], which achieves SOTA performance

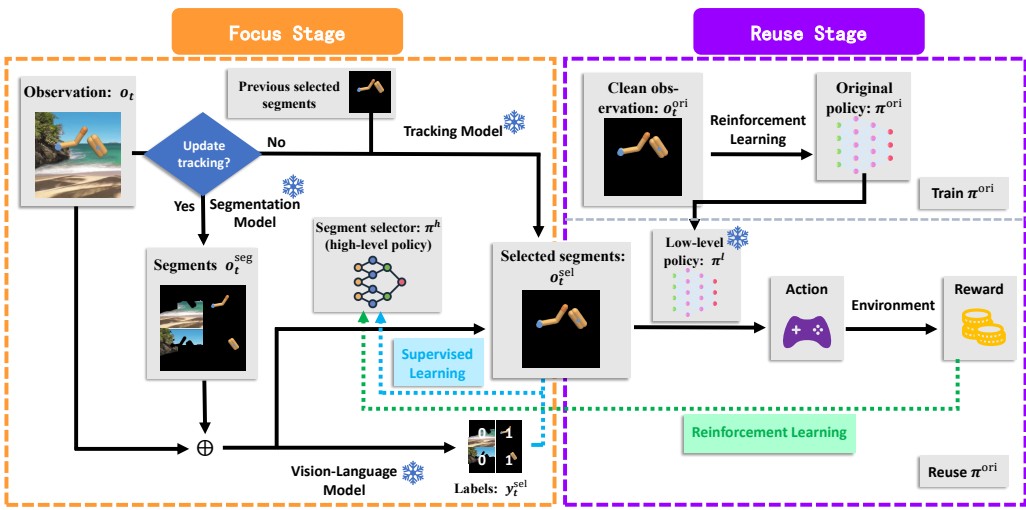

Figure 2: Architecture overview. FTR comprises a focus stage and a reuse stage, depicted in the orange and purple boxes, respectively. The focus stage utilizes a novel object selection mechanism to filter task-relevant segments, and the reuse stage applies a fixed, simulator-trained policy to generate actions based on the selected objects. The object selection mechanism consists of a trainable segment selector, a fixed segmentation model, and a fixed tracking model. The training of the segment selector synthesizes prior knowledge from a VLM and feedback from the environment.

in both tasks. In image segmentation task, the model takes an image $o \in \mathbb{R}^{C \times W \times H}$ ($C$, $W$, $H$ are respectively channels, width, and height) as input and outputs a set of $k$ binary masks $\left\{ m^i \mid i \in \{1, \cdots, k\}, m^i \in \{0,1\}^{W \times H} \right\}$. Each mask $m^i$ corresponds to a single object instance, with 1 indicating a pixel of the object. Segmented images are then obtained by element-wise multiplication ($\odot$), $\left\{ o_{\text{obj}}^i \mid i \in \{1, \cdots, k\}, o_{\text{obj}}^i = o \odot m^i \in \mathbb{R}^{C \times W \times H} \right\}$. In object tracking task, the model takes a video and masks as input and propagates the masks across the video. Given a video with $T$ frames $\left\{ o_t \mid t \in \{1, \cdots, T\}, o_t \in \mathbb{R}^{C \times W \times H} \right\}$ and $k$ object masks in the first frame $\left\{ m_1^i \mid i \in \{1, \cdots, k\}, m_1^i \in \{0,1\}^{W \times H} \right\}$, the goal of tracking is to predict the object masks in subsequent frames, $\left\{ m_t^i \mid t \in \{1, \cdots, T\}, i \in \{1, \cdots, k\} \right\}$.

## 4 Method

We present Focus-Then-Reuse (FTR), a hierarchical framework designed to quickly deploy policies in target domains with visual perturbations. FTR facilitates efficient policy adaptation by maintaining the core functionality of the original policy while dynamically compensating for visual perturbations through a learned "focus" module. In this section, we first provide a brief overview of the FTR framework. Following this, we introduce the forward process of our method, which is divided into two stages: the focus stage and the reuse stage. For the focus stage, we combine two filtering techniques to identify the task-relevant objects. For the reuse stage, we explain the training and reusing of the original policy. Finally, we present the training approach for the segment selector, which incorporates both supervised learning objective and reinforcement learning objective.

### 4.1 Architecture overview

An overview of FTR is provided in Fig. 2. FTR comprises two distinct stages: a high-level focus stage and a low-level reuse stage. They are in the orange and purple boxes, respectively. In the focus stage, an image observation with visual disturbance is processed. Based on whether the tracked objects require updating, task-relevant segments are obtained either through a segment selector (high-level policy $\pi^h$) or a tracking model. In the reuse stage, the selected objects $o_t^{\text{sel}}$ are fed into the

original policy, which is acquired via pre-training in a clean environment, to obtain the action. The training objective of $\pi^h$ combines RL with VLM supervision to enable efficient adaptation.

## 4.2 Focus stage: segmentation, selection and tracking

In this section, we introduce the focus stage in detail, as shown in the orange box in Fig. 2. At time step $t$, the image observation of the visually disturbed environment is denoted as $o_t \in \mathbb{R}^{C \times W \times H}$. The focus stage takes $o_t$ as input and outputs the filtered task-relevant objects, denoted as $o_t^{\text{sel}} \in \mathbb{R}^{C \times W \times H}$. Depending on whether the tracked objects need update (blue diamond box in Fig. 2), there are two approaches: The first approach uses a segmentation model and a segment selector (the selection pathway). The second approach employs a tracking model and historical selected images (the tracking pathway). We introduce these two pathways and explain their necessities.

### 4.2.1 Selection pathway

The selection pathway corresponds to the downward arrow labeled "Yes" in Fig. 2. This pathway filters objects through an image segmentation model and a segment selector $\pi^h$. First, an image segmentation model is employed to segment the disturbed observation $o_t$, yielding a set of $k$ segments: $o_t^{\text{seg}} = \left\{ o_t^{\text{seg}_i} \mid i \in \{1, \cdots, k\}, o_t^{\text{seg}_i} \in \mathbb{R}^{C \times W \times H} \right\}$. Subsequently, $\pi^h$ takes the $k$ segments and the original observation $o_t$ as input, and output the mean values $\boldsymbol{\mu}_t = (\mu_t^1, \mu_t^2, \cdots, \mu_t^k) \in (0,1)^k$. By sampling from a diagonal Gaussian distribution $\mathcal{N}(\boldsymbol{\mu}_t^\top, \sigma_h^2 \mathbf{I})$ with $\boldsymbol{\mu}_t$ as mean, hyperparameter $\sigma_h$ as variance ($\sigma_h = 0.1$ by default), and $\mathbf{I}$ as the identity matrix, we get action $a_t^{\text{sel}} = (a_t^{\text{sel}_1}, a_t^{\text{sel}_2}, \cdots, a_t^{\text{sel}_k})$. Using 0.5 as the threshold, the $i$-th segment is selected if its value $a_t^{\text{sel}_i} > 0.5$; otherwise, it is discarded. The selected segments are then integrated to form $o_t^{\text{sel}} \in \mathbb{R}^{C \times W \times H}$, representing the union of focused objects:

$$o_t^{\text{sel}} = \bigcup_{i \in \mathcal{I}_t} o_t^{\text{seg}_i}, \quad \text{where } \mathcal{I}_t = \left\{ i \mid i \in \{1, \cdots, k\}, a_t^{\text{sel}_i} > 0.5 \right\}. \tag{2}$$

By obtaining $o_t^{\text{sel}}$ through the above sampling method, we can effectively explore different segment selection patterns and prevent premature convergence to local optima.

### 4.2.2 Tracking pathway

While the selection pathway alone is sufficient to derive $o_t^{\text{sel}}$, this pathway suffers from a major issue: the inconsistency in segment selection. A segment selector that has not been sufficiently trained to develop a stable selection pattern is likely to produce results lacking short-term consistency, resulting in inconsistent actions and low-quality reward signals. Therefore, we propose another approach to derive $o_t^{\text{sel}}$, the tracking pathway. Whenever we obtain the selected objects in the selection pathway, we simultaneously record them in the tracking model. The tracking model then recognizes the selected objects $o_t^{\text{sel}}$ using previous selection results.

Till now, we have fully introduced the two pathways for selecting images. We propose a simple yet effective mechanism to combine these two pathways. We introduce a selection interval $T_{\text{sel}}$ ($T_{\text{sel}} = 20$ by default). If the current timestep $t$ is divisible by $T_{\text{sel}}$, the selection pathway is called; otherwise, the tracking pathway is used. In a corner case where no segment is selected by $\pi^h$, tracking must also be refreshed in the subsequent timestep, regardless of $T_{\text{sel}}$.

## 4.3 Reuse stage

Before adaptation, the original policy $\pi^{\text{ori}}$ is trained in a clean environment without visual perturbation using the DrQ-v2 algorithm [66]. In the reuse stage, $\pi^{\text{ori}}$ is copied and frozen as the lower-level policy $\pi^l$. As a common practice in visual RL, the last 3 frames of focused objects $o_{t-2}^{\text{sel}}, o_{t-1}^{\text{sel}}, o_t^{\text{sel}}$ are stacked and fed into $\pi^l$, yielding action $a_t$ for environment interaction to obtain reward $r_t$. Notably, the focus stage and the reuse stage are highly decoupled, meaning that our method is compatible with a range of RL algorithms. Additionally, our method requires no modifications to $\pi^l$.

## 4.4 Segment selector trained with supervised learning and RL

We introduce the training process of FTR in this section. During training, only the parameters of the segment selector $\pi^h$ are updated, while those of the segmentation model, tracking model, and low-level policy $\pi^l$ remain frozen. The training objective of the segment selector consists of two parts: the first is a supervised learning objective based on a VLM (blue dashed line in Fig. 2); the second is an RL objective based on environmental rewards (green dashed line in Fig. 2). Recalling the way humans handle complex environments, as mentioned in Section 1, supervised learning can be seen as leveraging prior knowledge, while RL resembles adjusting based on environmental feedback.

### 4.4.1 Supervised learning objective

We use the discrepancy between the VLM's selection and the output of $\pi^h$ as the supervised loss. Given segments $o_t^{\text{seg}}$, an example image from the source domain, and output format, the VLM returns $\mathbf{y}_t = (y_t^1, y_t^2, \cdots, y_t^k)$, where $y_t^i \in \{0, 1\}$, indicating whether each segment should be focused on. $(o_t, o_t^{\text{seg}}, \mathbf{y}_t)$ are added to $\mathcal{D}_{\text{SL}}$, the supervision dataset, for training. We provide more details about VLM in Appendix A.5.

With the selection result of VLM, an intuitive way of training is to use the Binary Cross Entropy (BCE) loss. However, we point out that the rigidity binarize of $\boldsymbol{\mu}_t$ will degrade the subsequent RL optimization. Instead, we use a "softer" margin-regularized loss function:

$$\mathcal{L}_{\text{SL}}(\delta) = \mathop{\mathbb{E}}_{\substack{(o, o^{\text{seg}}, \mathbf{y}) \sim \mathcal{D}_{\text{SL}} \\ a^{\text{sel}} \sim \pi^h(\cdot | o, o^{\text{seg}})}} \left[ \mathbf{y} \cdot \max(0, 0.5 + \delta - a^{\text{sel}}) + (\mathbf{1} - \mathbf{y}) \cdot \max(0, a^{\text{sel}} - (0.5 - \delta)) \right], \quad (3)$$

where $\delta$ is a predefined margin hyperparameter ($\delta = 0.1$ by default). For positive samples ($y^i = 1$), the loss penalizes predictions $a^{\text{sel}_i}$ that fall below $0.5 + \delta$ with linearly increasing loss. For negative samples ($y^i = 0$), it penalizes predictions exceeding $0.5 - \delta$.

### 4.4.2 RL objective

Although the VLM exhibits impressive reasoning capabilities, its predictions can be unreliable in certain scenarios, and its latency of several seconds is unacceptable for real-time control tasks. These limitations motivate the integration of RL to refine the policy. Specifically, $\pi^h$ is optimized using the PPO algorithm within a hierarchical framework. The state, action, and reward of $\pi^h$ are:

- **State** includes observation $o_t$ and segments $o_t^{\text{seg}}$ when $\pi^h$ is called on the selection pathway.
- **Action** is the selection vector $a_t^{\text{sel}}$ sampled from $\mathcal{N}(\boldsymbol{\mu}_t^\top, \sigma_h^2 \mathbf{I})$.
- **Reward** is the cumulative reward $\sum_{\tau=t}^{t+T_{\text{sel}}-1} r_\tau$ between two consecutive invocations of $\pi^h$.

Based on the MDP formulation, we train the high-level policy $\pi^h$ with the PPO algorithm, and the loss function $\mathcal{L}_{\text{RL}}$ is defined in Eq. 1. The RL objective relies on a simple yet valid assumption: the higher the accuracy of the segment selector, the greater the cumulative reward obtained by the low-level policy, and vice versa.

### 4.4.3 Combination of the objectives

Both objectives involve trade-offs: supervised learning enables rapid convergence but is limited by the VLM's accuracy and latency, whereas RL can achieve better performance but may result in slower training. The total loss function is defined as follows:

$$\mathcal{L} = \eta_{\text{SL}} \mathcal{L}_{\text{SL}} + \eta_{\text{RL}} \mathcal{L}_{\text{RL}}, \quad (4)$$

where the weights of $\mathcal{L}_{\text{SL}}$ and $\mathcal{L}_{\text{RL}}$ are $\eta_{\text{SL}}$ and $\eta_{\text{RL}}$, respectively. To complement the two objectives, we propose a dynamic adjustment mechanism of $\eta_{\text{SL}}$ and $\eta_{\text{RL}}$. This strategy allows the training to benefit from supervised learning for fast convergence in the early phase, while gradually shifting towards reinforcement learning for fine-tuning in the later phase. Specifically, when $t < T_1$, $\eta_{\text{SL}} = 1, \eta_{\text{RL}} = 0$. During $T_1 \leq t < T_2$, $\eta_{\text{SL}}$ decays linearly from 1 to 0 while $\eta_{\text{RL}}$ rises from 0 to 1, mitigating gradient misalignment between objectives. Note that the VLM is called only when $t < T_1$ with its outputs added to $\mathcal{D}_{\text{SL}}$. When $t > T_2$, $\eta_{\text{SL}} = 0, \eta_{\text{RL}} = 1$. By default, we set $T_1 = 5000$ and $T_2 = 10000$, thereby restricting VLM calls to $T_1/T_{\text{sel}} = 250$ times.

# 5 Experiments

In this section, we present the experimental results of FTR on 11 tasks, including 8 tasks of Deep-Mind Control Suite (DMC) [67] and 3 tasks of Franka Emika Robotics [68, 69]. DMC is a widely used benchmark of visual RL, while Franka Emika Robotics can better reflect real-world training scenarios. Finally, we conduct ablation studies to validate each module's effectiveness.

## 5.1 Experimental setup

We use 8 tasks from DMC: pendulum-swingup (ps), cartpole-swingup (cs), finger-spin (fs), hopper-stand (hs), hopper-hop (hh), cheetah-run (cr), walker-walk (ww), walker-run (wr), and 3 tasks for robotic manipulation: franka-reach (fr), franka-push (fp), and franka-door (fd). These tasks are diverse and comprehensive, ranging from single objects to multiple objects and from easy locomotion tasks to dexterous manipulation tasks.

Our research explores fast adaptation from source to target domains. The source domain corresponds to environments with no background. Policies are trained using the DrQ-v2 algorithm. We perform three independent runs and choose the policy with the best performance as the original policy $\pi^{\text{ori}}$. To simulate real-world visual disturbances, we use five diverse videos from the DMC-Generalization Benchmark [17], as shown in Fig. 3. These videos cover a range of indoor and outdoor scenes and serve as the backgrounds of five target domains. We perform domain adaptation independently on each target domain and report the average performance.

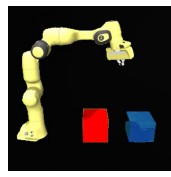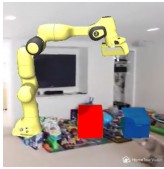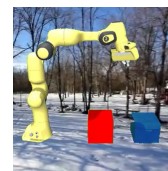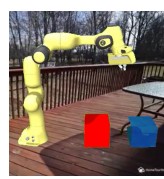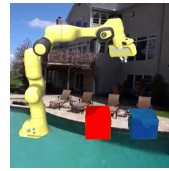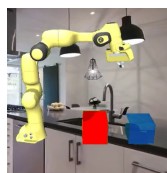

Figure 3: Source domain (leftmost) and five target domains of task franka-push.

We compare FTR with the following baseline methods:

- **DrQ-v2 (clean)** [66]. DrQ-v2 is a classic visual RL algorithm. It is also used as the default method by FTR for training the original policy in the source domain. DrQ-v2 (clean) can be regarded as the potential upper bound for FTR's domain adaptation performance.

- **SimGRL** [13]. SimGRL achieves SOTA performance in visual generalization RL, demonstrating effectiveness in test environments with video backgrounds.

- **PAD** [23]. PAD is a classical visual domain adaptation method in reinforcement learning, and has shown effectiveness in adapting to target domains with simple backgrounds.

- **$Q^2$-learning** [11]. $Q^2$-learning is a representation learning-based method for performing RL directly on complex visual inputs.

We also compare against **FTR w/o SL** and **FTR w/o RL**. FTR w/o SL refers to the variant of FTR trained without supervised learning, relying solely on the RL objective. Conversely, FTR w/o RL denotes the variant trained without RL.

FTR and the baseline methods can be categorized into four groups: classical visual RL method DrQ-v2, generalization method that requires no interaction with target domains, adaptation methods that involve limited interaction, and robust training method that is trained directly on the target domains. Although we present the results of all methods in the same table, we clarify their differences:

- **Classical visual RL method (DrQ-v2 (clean))**: We perform three independent runs in the source domain and report the best performance.

- **Generalization method (SimGRL)**: Policies are trained in the source domain for 500k steps across three runs. The trained policies are then evaluated in the five target domains.

- **Adaptation methods (FTR, PAD)**: We first conduct three training runs in the source domain and select the best-performing policy as the initial policy. This policy is then adapted to each of the five target domains using three different seeds for 200k steps.

- **Robust training method ($Q^2$-learning)**: Policies are trained directly in each target domain for 500k steps.

We report the mean and standard deviation of performance across the five target domains for all methods except DrQ-v2 (clean).

## 5.2 Experiment results

Table 1 and Fig. 4 show the performance for FTR and baseline methods. Table 1 records the final performance. The solid lines and shaded area in Fig. 4 correspond to the mean and variance. FTR achieves the best performance on 10 out of 11 tasks.

Compared to baselines, FTR demonstrates clear advantages in both sample efficiency and performance. In terms of sample efficiency, FTR leverages prior knowledge from the VLM to achieve strong initial performance and improves rapidly using environmental feedback, achieving convergence within 50k steps in most cases. Regarding final performance, FTR outperforms baselines on all tasks except cartpole-swingup. On average across all tasks, FTR retains over 85% of the potential upper bound represented by DrQ-v2 (clean). On the most complex robotic manipulation tasks, FTR shows considerable advantages over other methods. SimGRL, the SOTA visual generalization RL method, performs decently on most tasks, but compared to FTR, it only holds an advantage on the cartpole-swingup task, a task with sparse rewards and challenging object segmentation. This suggests that even with strong augmentation during training, encountering unseen disturbances at deployment can still lead to substantial performance degradation. Similarly, $Q^2$-learning can obtain a certain level of performance on most tasks. However, its performance remains inferior to FTR except on cartpole-swingup. This indicates that traditional representation learning methods still struggle to handle complex visual perturbations. Despite being a classic visual domain adaptation RL method, PAD performs poorly on all 11 tasks. This indicates that existing adaptation methods struggle with complex, near-real-world target domains with video background.

As an ablation study to evaluate the respective contributions of supervised learning and reinforcement learning in FTR, we compare FTR with its two variants: FTR w/o SL and FTR w/o RL. Comparing FTR and FTR w/o SL, we observe similar final performance; however, FTR converges significantly faster. This suggests that the prior knowledge from VLM offers a well-informed initialization. On the cartpole-swingup task, the performance of FTR w/o SL exhibits a notable performance drop. We speculate that this is due to the task's reward being sensitive to state variations, which causes excessive noise in the RL objective. Comparing FTR and FTR w/o RL, it can be observed that incorporating RL consistently improves performance. This indicates that the supervisory signal alone is insufficient, as errors from the VLM can lead to sub-optimal policies. Therefore, RL is essential for achieving optimal domain-adaptive performance.

Table 1: Performance comparison of FTR and baselines (mean $\pm$ std). Note that we train and evaluate DrQ-v2 in the clean source domain, and test all other methods in visually perturbed target environments. For DrQ-v2 (clean), we report the best performance over three runs as the potential upper bound for other methods.

| Task | DrQ-v2 (clean) | PAD | $Q^2$-learning | SimGRL | FTR w/o RL | FTR w/o SL | FTR (ours) |
|------|---------------|-----|----------------|--------|------------|------------|------------|
| ps | 829.0 | 0.5 $\pm$ 0.4 | 436.6 $\pm$ 331.2 | 46.1 $\pm$ 49.8 | 708.1 $\pm$ 145.5 | 770.9 $\pm$ 85.4 | **786.7** $\pm$ **82.3** |
| cs | 829.3 | 79.3 $\pm$ 8.9 | **807.3** $\pm$ **84.1** | 578.6 $\pm$ 306.1 | 541.2 $\pm$ 187.2 | 373.7 $\pm$ 152.1 | 646.0 $\pm$ 168.5 |
| fs | 973.3 | 0.8 $\pm$ 0.7 | 695.8 $\pm$ 121.8 | 351.6 $\pm$ 273.8 | 718.0 $\pm$ 183.8 | **909.1** $\pm$ **62.3** | 903.9 $\pm$ 71.2 |
| hs | 888.9 | 2.2 $\pm$ 2.2 | 264.2 $\pm$ 125.8 | 746.9 $\pm$ 139.3 | 563.0 $\pm$ 209.3 | 813.7 $\pm$ 68.0 | **825.5** $\pm$ **83.6** |
| hh | 336.4 | 0.5 $\pm$ 0.7 | 91.7 $\pm$ 48.9 | 124.2 $\pm$ 20.3 | 231.1 $\pm$ 77.2 | 300.9 $\pm$ 17.1 | **307.3** $\pm$ **20.8** |
| cr | 516.0 | 6.8 $\pm$ 6.8 | 221.6 $\pm$ 49.8 | 221.4 $\pm$ 110.6 | 344.9 $\pm$ 124.7 | 427.3 $\pm$ 70.5 | **446.1** $\pm$ **65.0** |
| ww | 957.1 | 26.3 $\pm$ 10.5 | 325.3 $\pm$ 62.2 | 734.0 $\pm$ 41.6 | 558.6 $\pm$ 212.2 | 891.5 $\pm$ 41.3 | **899.7** $\pm$ **45.4** |
| wr | 415.4 | 25.8 $\pm$ 12.4 | 141.5 $\pm$ 17.9 | 312.1 $\pm$ 27.8 | 244.3 $\pm$ 94.3 | 359.2 $\pm$ 36.7 | **366.5** $\pm$ **37.8** |
| fr | 948.3 | −1.1 $\pm$ 6.4 | 657.7 $\pm$ 220.0 | 10.3 $\pm$ 14.9 | 781.8 $\pm$ 161.9 | **873.3** $\pm$ **52.0** | 860.5 $\pm$ 70.6 |
| fp | 121.5 | 10.4 $\pm$ 11.6 | 19.0 $\pm$ 12.9 | 6.5 $\pm$ 4.2 | 64.2 $\pm$ 47.4 | **107.4** $\pm$ **15.1** | 96.8 $\pm$ 33.8 |
| fd | 156.8 | 0.3 $\pm$ 0.4 | 2.7 $\pm$ 4.9 | 91.5 $\pm$ 74.2 | 108.6 $\pm$ 37.2 | 122.7 $\pm$ 28.5 | **123.0** $\pm$ **34.9** |

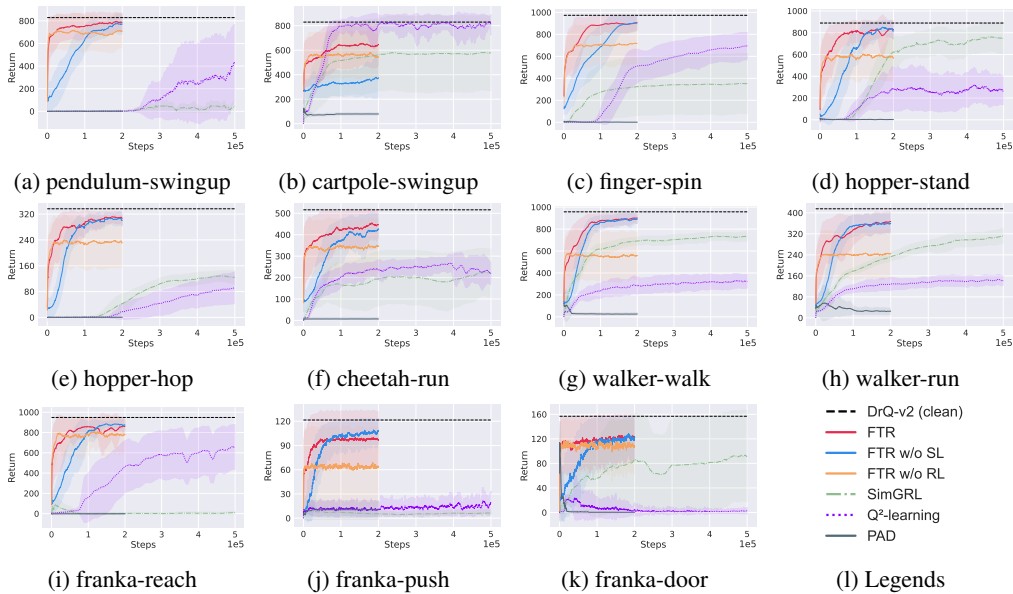

Figure 4: Training curves on DeepMind Control Suite (a-h) and Franka Emika Robotics (i-k). Note that for the adaptation methods (FTR, PAD), we set the adaptation duration to 200k steps, and for other methods, we train for 500k steps. FTR has converged by 100k steps across all tasks.

## 5.3 Further ablation studies

We conduct ablation studies on different selection intervals $T_{sel}$, SL-to-RL transition timesteps $T_1$, and supervised learning objectives $\mathcal{L}_{SL}$ on the finger-spin task.

The results under different values of $T_{sel}$ are in Fig. 5(a). When $T_{sel} = 1$ (using only the selection pathway while disabling the tracking pathway), the RL process fails to converge, leading to poor final performance. This highlights the critical role of the tracking pathway in FTR. As mentioned in Section 4.2.2, the tracking pathway helps maintain consistency in segment selection, thereby enhancing the stability in RL training. For $T_{sel} \in \{10, 20, 40\}$, performance exhibits minor variation.

In Fig. 5(b), the performance shows negligible differences between $T_1 = 1000$ and $T_1 = 5000$, demonstrating the effectiveness of our method even with limited supervision from the VLM. When $T_1 = 10000$, a slight performance drop is observed, which may be attributed to overfitting to $\mathcal{D}_{SL}$.

As shown in Fig. 5(c), using BCE loss as the supervised learning objective leads to a significant degradation in performance, especially during the transition from supervised learning to reinforcement learning. This underscores the importance of the proposed margin-regularized loss in Eq. 3 for ensuring a stable transition from supervised learning to reinforcement learning.

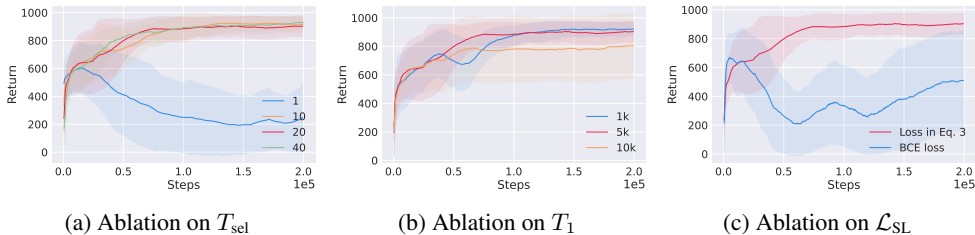

Figure 5: Ablation studies on different selection intervals $T_{sel}$, SL-to-RL transition timesteps $T_1$, and supervised learning objectives $\mathcal{L}_{SL}$ on the finger-spin task.

# 6 Conclusion

In this work, we propose a novel Focus-Then-Reuse (FTR) framework to achieve rapid policy deployment in real-world environments with background disturbances. The core of FTR lies in training a segment selector using both environmental rewards and VLM's supervision to identify task-relevant objects, while directly applying source domain policy on the filtered visual inputs. The focus stage and the reuse stage are highly decoupled, meaning that FTR is compatible with a range of generalization RL algorithms and has the potential to handle complex distribution shift. Furthermore, we propose a novel object selection mechanism that combines segmentation model and tracking model to improve object selection consistency and enhance the stability in RL training. Experimental results on the DeepMind Control Suite and Franka Emika Robotics indicate that our method effectively synthesizes prior knowledge from the VLM and environmental feedback, demonstrating advantages over baseline methods in performance, efficiency, and interpretability.

For future work, we suggest three aspects worth improving and exploring. First, while FTR demonstrates effective deployment in target domains with visual perturbation, it struggles in additional disturbances such as camera pose variations. A potential solution involves incorporating visual generalization RL approaches to enhance the robustness of the source domain policy. Second, the fixed selection interval $T_{sel}$ relies on manual configuration. Developing an adaptive scheduler for $T_{sel}$ could potentially enhance performance. Finally, the performance of FTR depends on the segmentation and tracking model, suggesting the need for foundation models specifically tailored for downstream RL tasks.

## Acknowledgments

We thank the reviewers for their insightful and valuable comments. This work is supported by the National Science Foundation of China (62276126, 62250069).

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

# Appendix

## A Implementation Details

In this section, we describe the implementation details of our work.

### A.1 FTR implementation

The source code is available at `https://github.com/LAMDA-RL/FTR`. The code is modified from DMC-Generalization Benchmark [17] and FTD [10]. The PPO algorithm used in FTR is implemented based on `https://iclr-blog-track.github.io/2022/03/25/ppo-implementation-details/`. The DrQ-v2 algorithm is implemented based on `https://github.com/facebookresearch/drqv2`. For details on hyperparameters and network architecture, please refer to the "Hyperparameters" section and the "Network Architecture" section. Algorithm for FTR is shown in Algorithm 1. Most experiments are conducted on a server outfitted with 2 AMD EPYC 7542 32-Core Processor CPUs, 504GB of RAM, and 8 GPUs, each with a performance of over 35 TFLOPS, running Ubuntu 22.04. Training in the source domain using the DrQ-v2 algorithm for 500k steps takes about 1 day. Adapting in the target domain for 200k steps takes about 8 hours.

---

**Algorithm 1** Focus-Then-Reuse

---

1: **Initialize:** segmentation model, tracking model, VLM, segment selector $\pi^h$, original policy trained in clean environment $\pi^{\text{ori}}$, replay buffer $\mathcal{D}_{\text{RL}}$, VLM supervision dataset $\mathcal{D}_{\text{SL}}$
2: **for** $t = 1$ to $N$ **do**
3:     **if** $t \% T_{\text{sel}} == 0$ **then**
4:         use the segmentation model to generate $o_t^{\text{seg}}$ from $o_t$
5:         generate high-level action $a_t^{\text{sel}} = \pi^h(o_t, o_t^{\text{seg}})$
6:         select $o_t^{\text{sel}}$ according to $a_t^{\text{sel}}$
7:         initialize the tracking model with $o_t^{\text{sel}}$ and $o_t$
8:         **if** $t < T_1$ **then**
9:             gain VLM's supervision information $\mathbf{y}_t$
10:             add $\langle o_t, o_t^{\text{seg}}, \mathbf{y}_t \rangle$ to $\mathcal{D}_{\text{SL}}$
11:         **end if**
12:         add $\langle \{o_{t-T_{\text{sel}}}, o_{t-T_{\text{sel}}}^{\text{seg}}\}, \{o_t, o_t^{\text{seg}}\}, a_{t-T_{\text{sel}}}^{\text{sel}}, \sum_{\tau=t-T_{\text{sel}}}^{t-1} r_\tau \rangle$ to $\mathcal{D}_{\text{RL}}$
13:     **else**
14:         use the tracking model to recognize $o_t^{\text{sel}}$ from $o_t$
15:     **end if**
16:     input $(o_{t-2}^{\text{sel}}, o_{t-1}^{\text{sel}}, o_t^{\text{sel}})$ into $\pi^{\text{ori}}$ to obtain $a_t$
17:     take action $a_t$ in the environment to receive the next observation $o_{t+1}$ and reward $r_t$
18:     sample a batch $\mathcal{B}_{\text{SL}}$ from $\mathcal{D}_{\text{SL}}$ and calculate $\mathcal{L}_{\text{SL}}$ according to Eq. 3
19:     sample a batch $\mathcal{B}_{\text{RL}}$ from $\mathcal{D}_{\text{RL}}$ and calculate $\mathcal{L}_{\text{RL}}$ according to Eq. 1
20:     calculate $\mathcal{L}$ according to Eq. 4 and update $\pi^h$
21:     update $\eta_{\text{SL}}$ and $\eta_{\text{RL}}$
22: **end for**

---

Segment Anything Model 2 (SAM 2) serves as the default segmentation model and tracking model in FTR. The official implementation of SAM 2 only supports offline video object tracking, not live streaming video. Therefore, we utilize the implementation from `https://github.com/Gy920/segment-anything-2-real-time`, which makes SAM 2 possible for real-time video applications. To improve training speed and memory efficiency, we align with FTD's preprocessing [10] to heuristically filter the segmentation model's outputs into $k = 9$ instances.

The default VLM used in FTR is Qwen-VL-Max [33]. For more details, please refer to the "VLM Details" section.

## A.2 Baseline implementation

SimGRL and PAD are implemented following the open-source code, Table 2 includes the links of them. $Q^2$-learning is implemented according to the original paper, and is included in our code.

Table 2: Links to the open-source code of baseline methods.

| Method | Open-Source URL |
|---|---|
| SimGRL | `https://github.com/W-Song11/SimGRL-Code` |
| PAD | `https://github.com/nicklashansen/dmcontrol-generalization-benchmark` |

## A.3 Hyperparameters

Table 3 shows the hyperparameters for reproducing the experiments.

Table 3: Hyperparameters.

| Hyperparameters of environments | |
|---|---|
| frame size | $168 \times 168$ (franka-push, franka-door), $84 \times 84$ (otherwise) |
| frame stack | 3 |
| episode length | 200 (franka-push, franka-door), 1000 (otherwise) |
| action repeat | 2 (finger-spin, pendulum-swingup), 4 (otherwise) |
| Hyperparameters of DrQ-v2 | |
| train steps | $5 \times 10^5$ |
| replay buffer size | $1 \times 10^5$ |
| exploration steps | $1 \times 10^4$ |
| $n$-step returns | 3 |
| batch size | 256 |
| optimizer | Adam |
| actor & critic learning rate | $1 \times 10^{-4}$ |
| discount factor | 0.99 |
| critic Q-function soft-update rate $\tau$ | 0.01 |
| exploration stddev. clip | 0.3 |
| exploration stddev. schedule | linear(1.0,0.1,100000) |
| Hyperparameters of focus stage | |
| SAM 2 checkpoint | sam2_hiera_tiny |
| adapt steps | $2 \times 10^5$ |
| number of segments $k$ | 9 |
| selection interval $T_{\text{sel}}$ | 20 |
| SL-to-RL transition timestep $T_1$ | 5000 |
| transition end timestep $T_2$ | 10000 |
| policy stddev. $\sigma_h$ | 0.1 |
| optimizer | Adam |
| batch size | 128 |
| learning rate | $3 \times 10^{-4}$ |
| clip ratio of PPO | 0.2 |
| discount factor | 0.5 |
| GAE lambda | 0.95 |
| $\mathcal{L}_{\text{SL}}$ objective margin $\delta$ | 0.1 |

## A.4 Network architecture

Below are the network architectures of the main components of FTR. Here, `MLP(n)` denotes a fully-connected layer with output size of $n$; `LayerNorm()` denotes applying layer normalization; `Conv2D(c, k, s, p)` denotes a 2D convolution layer of output channel $c$, kernel size $k$, stride $s$, and padding $p$; `Maxpool2D(k, s)` denotes a 2D max-pooling layer of kernel size $k$ and stride $s$; `Flatten()` denotes a flatten layer; `ReLU()` denotes a rectified linear unit; `Tanh()` denotes a hyperbolic tangent function; $\{\cdots\} \times k$ denotes repeating the layers within brace for $k$ times.

### A.4.1 Network architecture of the original policy

The original policy $\pi^{\text{ori}}$ is trained in the clean environment without visual perturbation using the DrQ-v2 algorithm. The actor and critic share the same image encoder.

**Encoder:**

```
Conv2D(32, 3, 2, 0) => ReLU() => {Conv2D(32, 3, 1, 0) => ReLU()}×3 =>
Flatten()
```

**Actor:**

```
MLP(50) => LayerNorm() => Tanh() => {MLP(1024) => ReLU()}×2 =>
MLP(action_dim)
```

**Critic:**

```
MLP(50) => LayerNorm() => Tanh() => {MLP(1024) => ReLU()}×2 => MLP(1)
```

### A.4.2 Network architecture of the segment selector

The segment selector (high-level policy $\pi^h$) is trained with the PPO algorithm. The actor $\pi^h$ and value function $V$ share the same embedding module $\phi$.

**Embedding module $\phi$:**

```
Conv2D(32,3,2,1) => {ReLU() => Conv2D(32,3,1,1) => Maxpool2D(2,2)}×4 =>
Flatten() => MLP(128)
```

**Actor $\pi^h$:**

$\pi^h$ adopts an attention-like mechanism to capture the relationship between $o_t^{\text{seg}}$ and $o_t$. First, the inputs are transformed into latent representation $\phi(o_t) \in \mathbb{R}^{1 \times D}$ and $\phi(o_t^{\text{seg}}) \in \mathbb{R}^{k \times D}$ through an embedding module $\phi$, where $D$ is the dimension of the latent space. Next, $\phi(o_t)$ undergoes linear projection to generate query vector $\mathbf{q}_t \in \mathbb{R}^{1 \times D}$, while $\phi(o_t^{\text{seg}})$ are mapped to key vectors $\mathbf{k}_t \in \mathbb{R}^{k \times D}$. Scaled dot-product scores are computed between $\mathbf{q}_t$ and $\mathbf{k}_t$:

$$\text{score}_t = \frac{\mathbf{q}_t \mathbf{k}_t^\top}{\sqrt{D}} \in \mathbb{R}^{1 \times k}. \tag{5}$$

Linear projection $W_i^k$ and $W_i^q, (i = 1, 2, 3, 4)$: `MLP(128)`

The scores can be interpreted as the relevance of $k$ segments to the task, where a higher value indicates stronger task relevance of the corresponding segments. Then, a sigmoid function is used to generate the probability of sampling each segment:

$$\boldsymbol{\mu}_t = (\mu_t^1, \mu_t^2, \cdots, \mu_t^k) = \text{Sigmoid}(\text{score}_t) \in (0, 1)^k. \tag{6}$$

**Value $V$:**

$V$ adopts an attention mechanism. Given latent representation $\phi(o_t)$ and $\phi(o_t^{\text{seg}})$ through the embedding module $\phi$, $\phi(o_t)$ undergoes linear projection to generate query vector, while $\phi(o_t^{\text{seg}})$ are mapped to key vectors and value vectors.

Linear projection $W_i^k$ and $W_i^q, (i = 1, 2, 3, 4), W^v$: `MLP(128)`

The outputs of the attention mechanism are fed into a value head to get the value of current state.

Value head: `MLP(1)`

### A.5 VLM details

We use Qwen-VL-Max [33] by default. Given segments $o_t^{\text{seg}}$ and prompt, the VLM returns $\mathbf{y}_t = (y_t^1, y_t^2, \cdots, y_t^k), y_t^i \in \{0, 1\}$, indicating whether each segment should be focused on. The prompt template is shown in Fig. 6, including an example image from source domain, segments $o_t^{\text{seg}}$ and output format. The example images for each task used in the prompt are shown in Fig. 7. The total cost of API calls is below \$100 for our experiment. The average response time for API calls is

approximately 7 seconds. The number of VLM calls for a single run in our experiment is $T_1/T_{\text{sel}} = 5000/20 = 250$.

Task: Determine if the object in each of the candidate images 1-9 is part of the *articulated* object(s) in the target image. Note that the articulated object may be in *different poses* or joint configurations across images. Output the results in JSON format.

Target Image: [Image Placeholder]
Candidate Image 1: [Image Placeholder]
Candidate Image 2: [Image Placeholder]
Candidate Image 3: [Image Placeholder]
Candidate Image 4: [Image Placeholder]
Candidate Image 5: [Image Placeholder]
Candidate Image 6: [Image Placeholder]
Candidate Image 7: [Image Placeholder]
Candidate Image 8: [Image Placeholder]
Candidate Image 9: [Image Placeholder]

Please return the results in JSON format as an array of objects. **The order of objects in the array must correspond to the order of candidate images provided (from 1 to 9).** Each object should contain the following fields:
- "image_id": Candidate image number (1-9)
- - "is_same_object": Boolean value (true if same object, false if different)

JSON Output Template:
```json[
{ "image_id": "1", "is_same_object": boolean },
{ "image_id": "2", "is_same_object": boolean },
{ "image_id": "3", "is_same_object": boolean },
 { "image_id": "4", "is_same_object": boolean },
{ "image_id": "5", "is_same_object": boolean },
{ "image_id": "6", "is_same_object": boolean },
{ "image_id": "7", "is_same_object": boolean },
{ "image_id": "8", "is_same_object": boolean },
{ "image_id": "9", "is_same_object": boolean }]```

Figure 6: VLM prompt template.

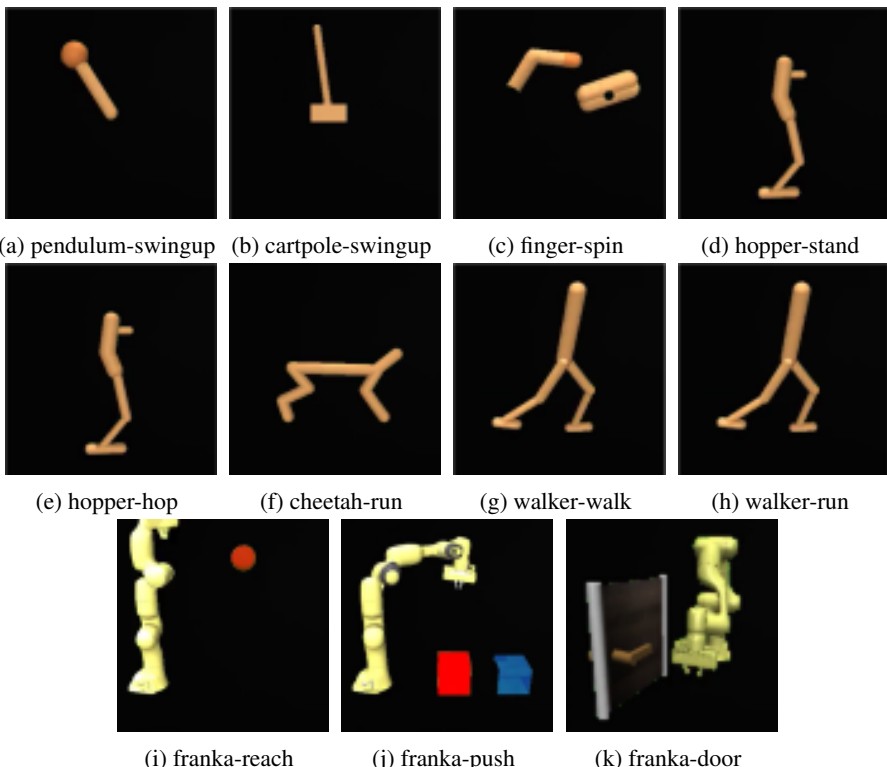

(a) pendulum-swingup  (b) cartpole-swingup  (c) finger-spin  (d) hopper-stand

(e) hopper-hop  (f) cheetah-run  (g) walker-walk  (h) walker-run

(i) franka-reach  (j) franka-push  (k) franka-door

Figure 7: Source domain of the DeepMind Control Suite (a-h) and Franka Emika Robotics (i-k) tasks.

# B Additional Experimental Results

## B.1 Further ablation studies

We conduct ablation studies on different selection intervals $T_{sel}$, SL-to-RL transition timesteps $T_1$, and supervised learning objectives $\mathcal{L}_{SL}$ on the finger-spin task. Results are shown in Table 4, Table 5, Table 6, and Fig. 5.

Table 4: Ablation on $T_{sel}$ on the finger-spin task.

| $T_{sel}$ | Performance (mean $\pm$ std) |
|---|---|
| 1 | $249.6 \pm 244.0$ |
| 10 | $915.7 \pm 55.9$ |
| 20 | $903.9 \pm 71.2$ |
| 40 | $928.9 \pm 47.8$ |

Table 5: Ablation on $T_1$ on the finger-spin task.

| $T_1$ | Performance (mean $\pm$ std) |
|---|---|
| 1000 | $922.5 \pm 44.4$ |
| 5000 | $903.9 \pm 71.2$ |
| 10000 | $802.8 \pm 229.5$ |

Table 6: Ablation on $\mathcal{L}_{SL}$ on the finger-spin task.

| $\mathcal{L}_{SL}$ | Performance (mean $\pm$ std) |
|---|---|
| The proposed loss in Eq. 3 | $903.9 \pm 71.2$ |
| Binary Cross Entropy (BCE) loss | $510.4 \pm 337.8$ |

## B.2 Performance of the baselines in the clean environments

Table 7 shows the results of the baselines in the clean environment. The experiments are conducted with three random seeds. PAD and SimGRL perform well on most tasks, while $Q^2$-learning performs decently except in franka-push and franka-door.

Table 7: Performance of the baselines in the clean environments (mean $\pm$ std).

| Task | PAD | $Q^2$-learning | SimGRL |
|---|---|---|---|
| pendulum-swingup | $665.0 \pm 437.7$ | $912.0 \pm 32.0$ | $910.0 \pm 6.0$ |
| cartpole-swingup | $859.6 \pm 14.5$ | $874.2 \pm 0.7$ | $862.8 \pm 7.2$ |
| finger-spin | $908.5 \pm 18.5$ | $653.2 \pm 262.9$ | $979.5 \pm 6.4$ |
| hopper-stand | $814.7 \pm 25.6$ | $845.0 \pm 18.3$ | $866.2 \pm 5.8$ |
| hopper-hop | $114.6 \pm 5.3$ | $170.7 \pm 14.0$ | $150.9 \pm 5.2$ |
| cheetah-run | $343.2 \pm 36.8$ | $385.5 \pm 13.3$ | $318.8 \pm 4.6$ |
| walker-walk | $915.4 \pm 20.4$ | $616.9 \pm 43.9$ | $879.1 \pm 43.9$ |
| walker-run | $318.3 \pm 7.3$ | $251.2 \pm 37.3$ | $349.4 \pm 11.3$ |
| franka-reach | $942.7 \pm 3.3$ | $884.2 \pm 44.6$ | $932.7 \pm 22.4$ |
| franka-push | $102.8 \pm 2.1$ | $54.8 \pm 5.6$ | $95.3 \pm 11.2$ |
| franka-door | $152.4 \pm 4.3$ | $20.4 \pm 0.3$ | $163.3 \pm 0.8$ |

## B.3 VLM accuracy

We calculate the accuracy of Qwen-VL-Max across different tasks in Table 8. The VLM's judgment is considered accurate when it selects task-related segments and refrains from selecting task-irrelevant ones.

Table 8: Accuracy of Qwen-VL-Max.

| Task | Accuracy (%) |
|------|--------------|
| pendulum-swingup | 97.84 |
| cartpole-swingup | 88.67 |
| finger-spin | 86.40 |
| hopper-stand | 94.23 |
| hopper-hop | 94.33 |
| cheetah-run | 96.32 |
| walker-walk | 89.77 |
| walker-run | 89.11 |
| franka-reach | 89.04 |
| franka-push | 85.52 |
| franka-door | 91.41 |

## B.4 Relationship between segment selection accuracy and reward

We conduct supplementary experiments on the franka-reach task to validate the relationship between segment selection accuracy and cumulative reward in an episode over five random seeds in Table 9. The results in the table demonstrate a clear correlation: When the segment selector accurately focuses on task-relevant objects, $\pi_l$ achieves the highest rewards. Conversely, incorrect selection leads to substantially lower rewards.

Table 9: Relationship between segment selection accuracy and performance (mean $\pm$ std).

| segment selection | Performance (mean $\pm$ std) |
|-------------------|------------------------------|
| Only task-relevant objects | $892.4 \pm 22.9$ |
| Task-relevant objects + 1 irrelevant object | $570.9 \pm 223.1$ |
| Task-relevant objects + 2 irrelevant objects | $495.3 \pm 259.3$ |
| All objects | $234.4 \pm 282.9$ |
| All task-irrelevant objects | $9.6 \pm 17.2$ |

## B.5 Experiments on more distractions

### B.5.1 Task-similar objects

To verify FTR's ability to handle task-similar objects in the target domain that could easily be misidentified, we introduce two scenarios: another cube in the background and another uncontrolled robotic arm in the background. We conduct experiments on the franka-reach task over three random seeds. The results, shown in Table 10, indicate that even with such objects in the background, FTR can still filter out task-relevant objects using environmental reward, thereby maintaining high performance.

Table 10: Performance of FTR on the franka-reach task in face of distracting objects (mean $\pm$ std).

| Another cube | Another uncontrolled robotic arm |
|--------------|----------------------------------|
| $913.0 \pm 18.5$ | $871.8 \pm 38.6$ |

### B.5.2 Illumination variation

We evaluate FTR's illumination generalization on the franka-reach task over three random seeds. To enhance the adapted policy's performance, as discussed in the conclusion of our paper, we introduce illumination perturbation during low-level policy $\pi_l$ training to enhance robustness. The results are in Table 11.

Row 1 shows performance without illumination-robust training, while Row 2 demonstrates FTR's adaptation under both illumination and background perturbations, maintaining the performance of $\pi_l$ despite illumination sensitivity. We introduce varying directions of illumination perturbations

during the training of $\pi_l$ in the source domain to enhance its robustness. As presented in Row 3, $\pi_l$'s performance improved significantly. Row 4 shows FTR's adaptation performance under both illumination and background perturbations, demonstrates that **FTR consistently maintained the performance of** $\pi_l$. Furthermore, benefiting from the enhanced robustness of $\pi_l$, the performance showed a notable improvement compared to Row 2.

Table 11: Performance of FTR in face of illumination variation (mean $\pm$ std).

| With background perturbation? | $\pi_l$ robust to illumination variation? | With FTR adaptation? | Performance (mean $\pm$ std) |
|---|---|---|---|
| no | no | no | $659.3 \pm 52.5$ |
| yes | no | yes | $599.1 \pm 62.1$ |
| no | yes | no | $952.3 \pm\ \ 4.1$ |
| yes | yes | yes | $929.5 \pm 11.6$ |

### B.6 Fine-tuning $\pi_l$ after adaptation

When $\pi_l$ lacks robustness or is deployed in an unforeseen perturbed environment, we show fine-tuning $\pi_l$ after adaptation (while fixing the selector) can further improve performance.

To demonstrate this, we conduct experiments on the franka-reach task over three random seeds and introduce two types of target domain perturbations: a $15°$ rotation of the camera around the z-axis and a $15°$ horizontal inclination of the robot arm's base. After training the selector for 200k steps, we fix its parameters and unfreeze $\pi_l$'s parameters to fine-tune $\pi_l$ for 50k steps using environmental reward. The results are in Table 12. The adapted FTR performance initially degrades. Nevertheless, after the 50k-step fine-tuning, the performance recovers to a commendable level.

Table 12: Performance after adaptation and fine-tuning $\pi_l$ (mean $\pm$ std).

| | After FTR adaptation | After fine-tuing $\pi_l$ |
|---|---|---|
| Camera rotation | $733.3 \pm 15.7$ | $904.6 \pm\ \ 8.7$ |
| Base inclination | $716.2 \pm 51.0$ | $906.0 \pm 13.7$ |

We posit that the reason for the effectiveness of the "adapt + fine-tune" process and the performance maintenance with non-robust $\pi_l$ is the same: The selector's RL training does not impose stringent requirements on the optimality of $\pi_l$. The selector can be guided towards correct learning as long as $\pi_l$ satisfies a key condition: the reward for a correctly chosen action by the selector is greater than that for an incorrectly chosen one. As shown in the experiment of Appendix B.4, this condition can be easily satisfied. Once the selector learns the correct selection patterns, fine-tuning $\pi_l$ based on the filtered images becomes highly efficient.

## C Visualization

We visualize $o_t$, $o_t^{\text{sel}}$, $o_t^{\text{seg}}$, and the selection result of $o_t^{\text{seg}}$ across all tasks (Figs. 8 to 18).

We show images from $t = 0$ to $t = 36$, sampled every 4 time steps. Recall that we set selection interval $T_{\text{sel}} = 20$. When $t = 0$ and $t = 20$, the selection pathway is called; otherwise, the tracking pathway is used. In the selection pathway, we displays the segments $o_t^{\text{seg}}, t \in \{0, 20\}$, and their selection probabilities. Here, a segment is selected if its probability $> 0.5$. The selected segments are then integrated to form $o_t^{\text{sel}}, t \in \{0, 20\}$, representing the union of focused objects. Whenever we obtain the selected objects $o_t^{\text{sel}}, t \in \{0, 20\}$ in the selection pathway, we simultaneously record them in the tracking model. In the tracking pathway, the tracking model recognizes the selected objects $o_t^{\text{sel}}$ using observation $o_t$.

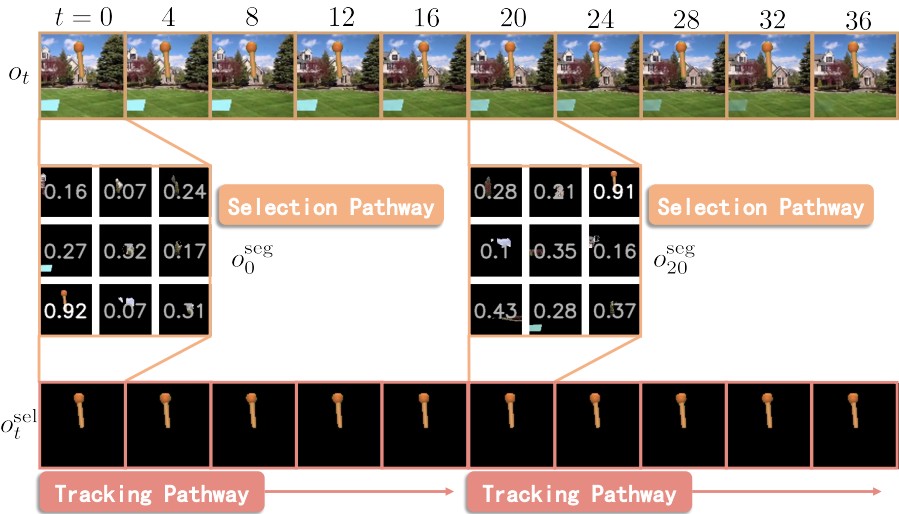

Figure 8: Visualization of pendulum-swingup.

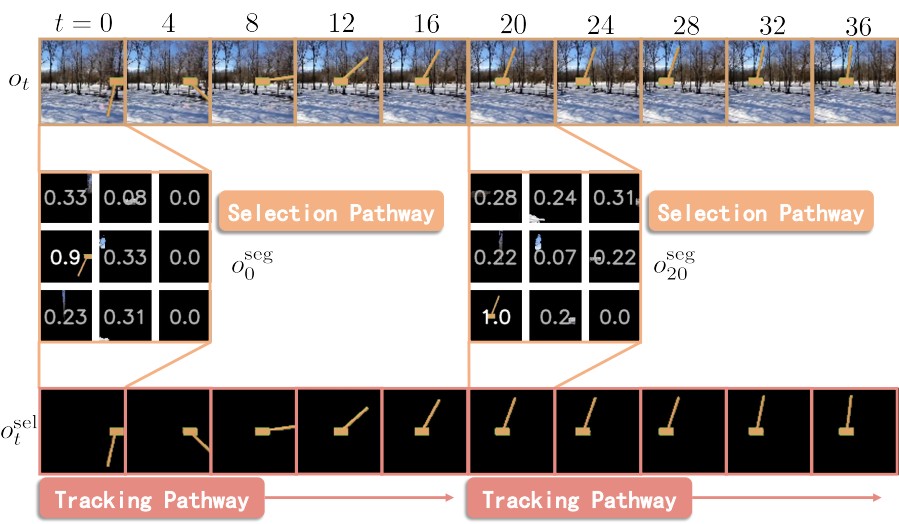

Figure 9: Visualization of cartpole-swingup.

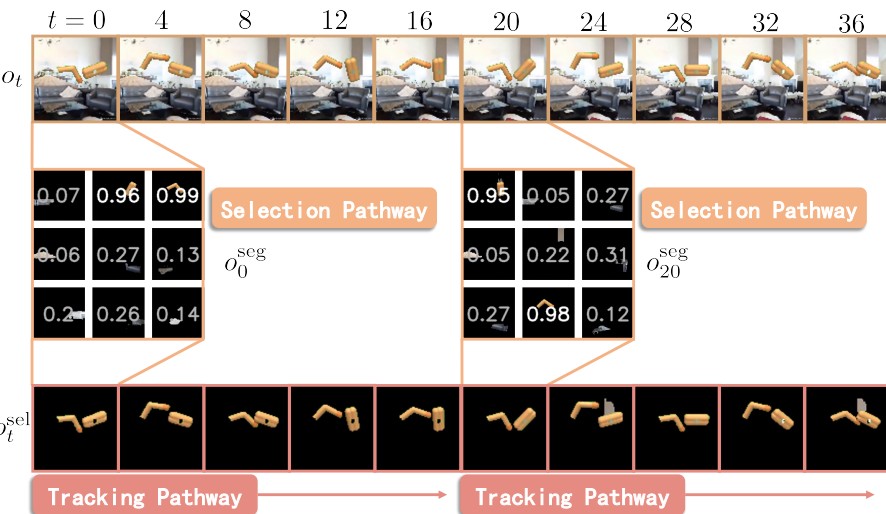

Figure 10: Visualization of finger-spin.

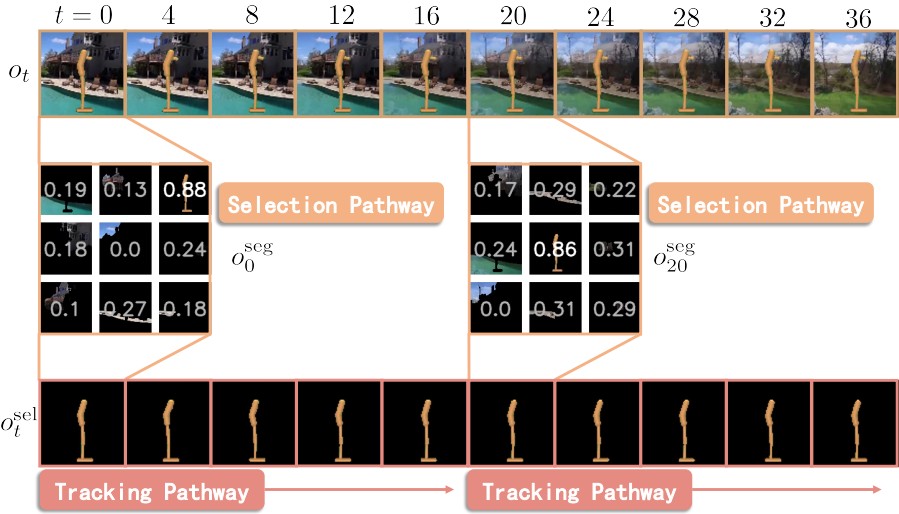

Figure 11: Visualization of hopper-stand.

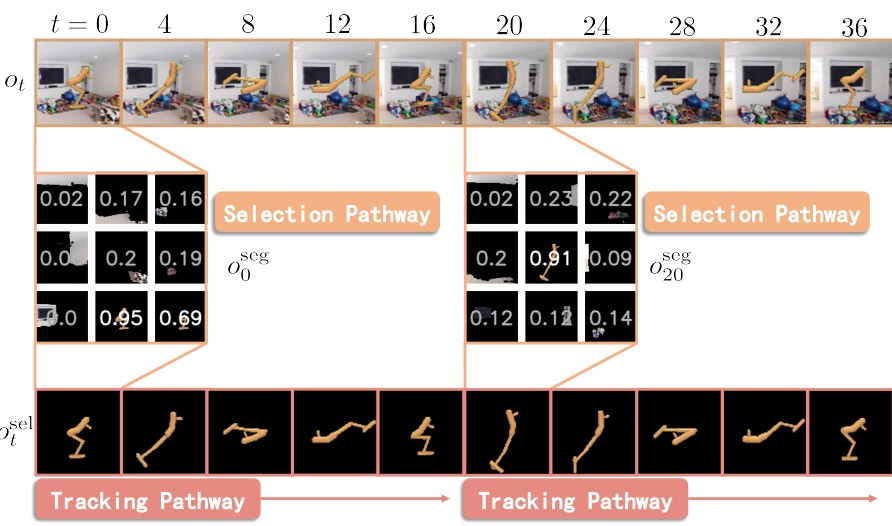

Figure 12: Visualization of hopper-hop.

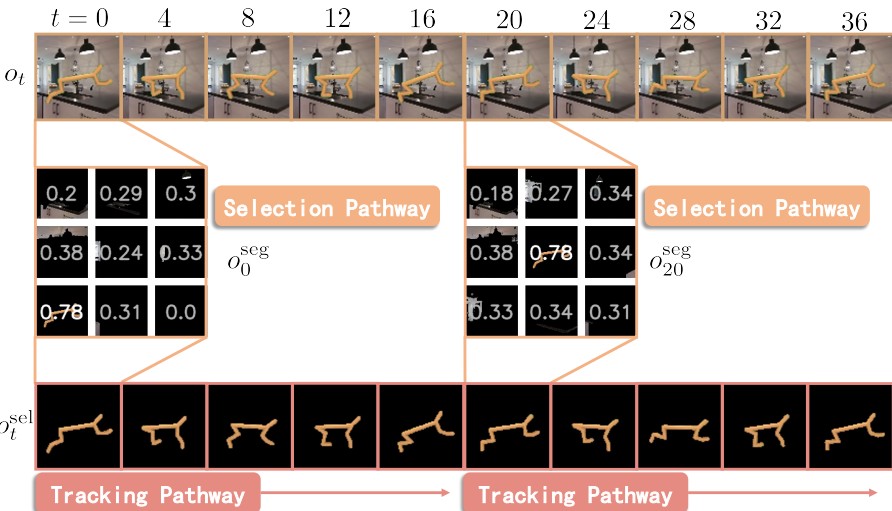

Figure 13: Visualization of cheetah-run.

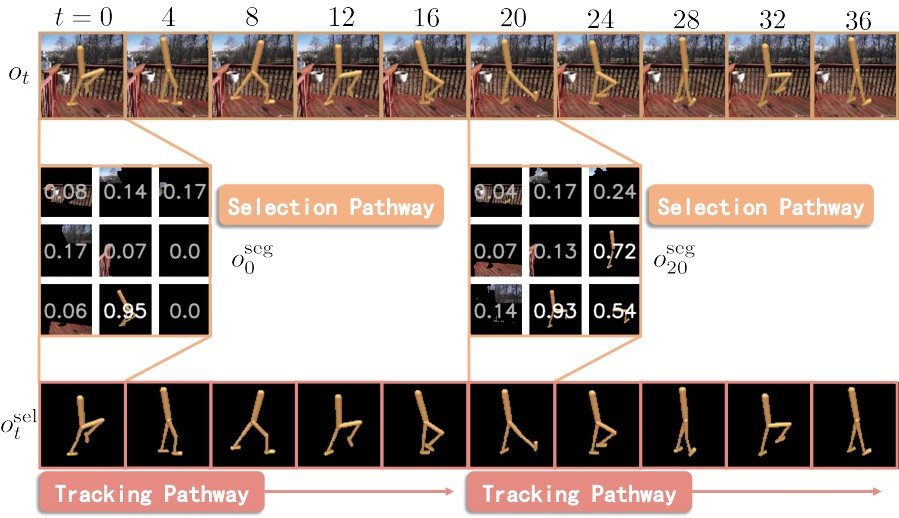

Figure 14: Visualization of walker-walk.

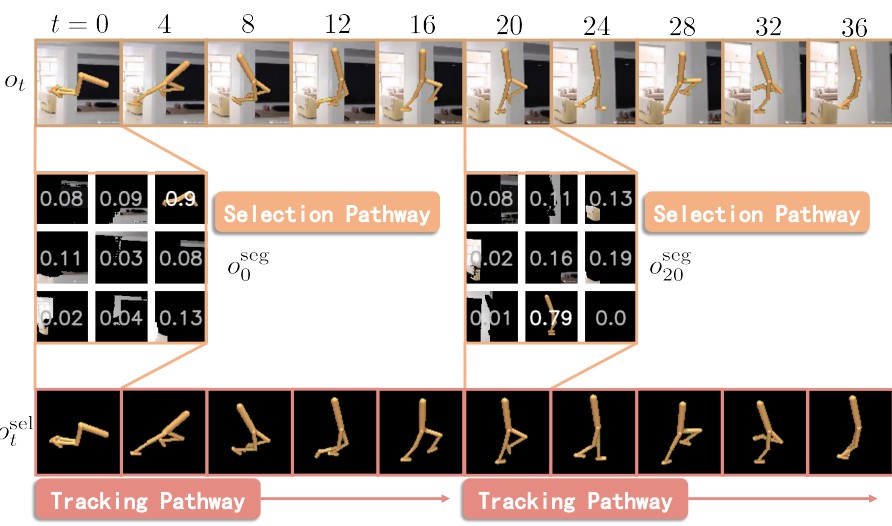

Figure 15: Visualization of walker-run.

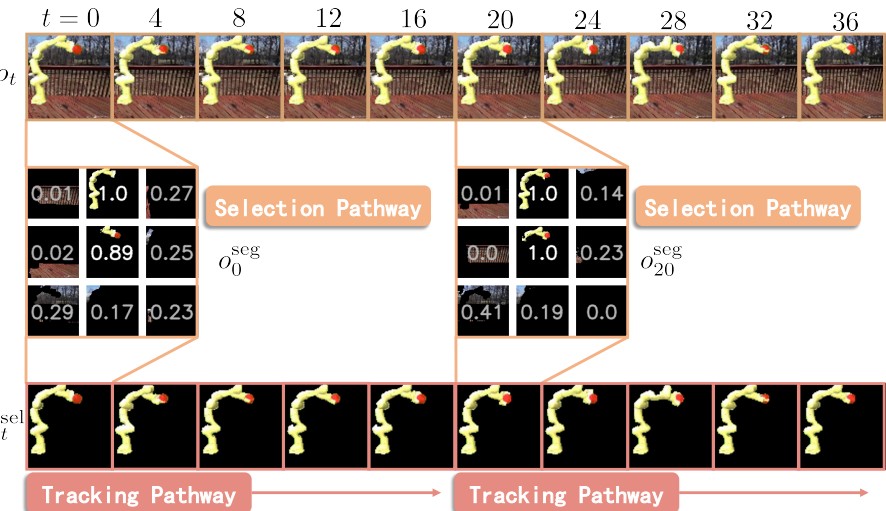

Figure 16: Visualization of franka-reach.

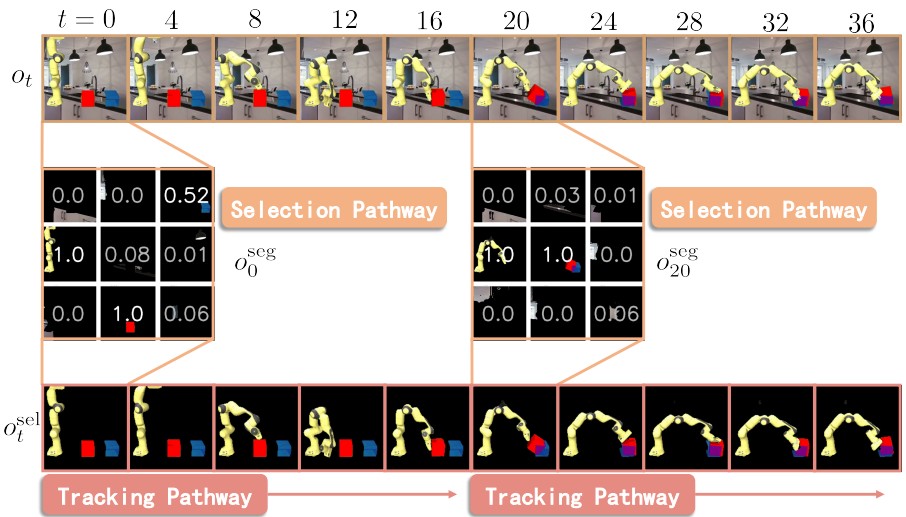

Figure 17: Visualization of franka-push.

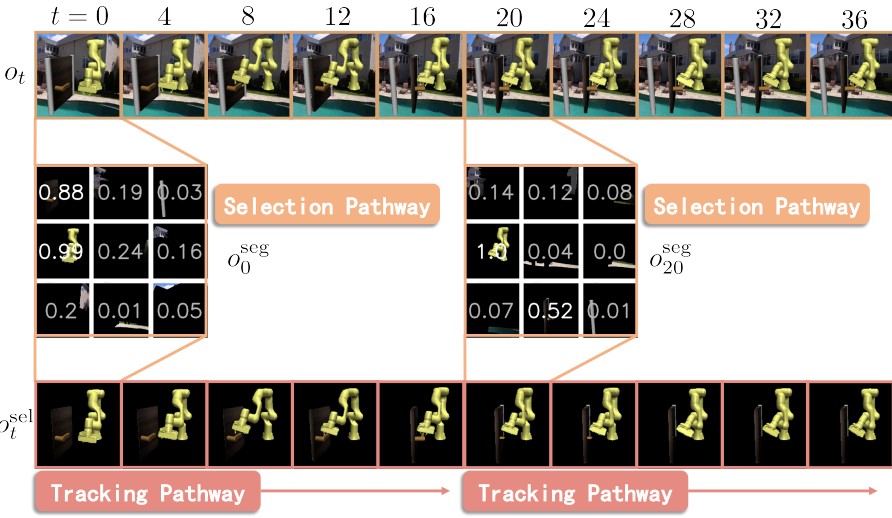

Figure 18: Visualization of franka-door.

