# OpenReview forum: "Focus-Then-Reuse: Fast Adaptation in Visual Perturbation Environments"
_NeurIPS.cc/2025/Conference — NeurIPS 2025 poster_

### Official Review · Reviewer_R2gz · 2025-06-13

**Clarity:** 2
**Significance:** 2
**Originality:** 2
**Rating:** 4
**Confidence:** 4

**Summary:**

The paper presents an approach for fast policy adaptation in environments with noisy/varying backgrounds. The approach uses a combination of image segmentation and tracking to extract only the information relevant for the task, while ignoring the uncontrollable components in the background. The representation extraction, in particular the image segmentation selection, is trained using a combination of supervised and RL training. The supervised signal comes from a VLM model that is used to get a high level scene understanding.

**Questions:**

While I think the idea of the paper is interesting, I have concern around the complexity of the approach and scalability.

- Why is it necessary to have both a tracking and segmentation model given that they are used in an exclusive way? Is it related to cost trade-off for the usage of the VLM?
- The fact that you need to train the frozen policy using no background seems quite a big limitation to me. Would it be possible to remove such constraint?
- How would you scale your approach to more complicated setting, eg navigation in Habit, robocasa, ManiSkill-HAB, Carla. In this case it seems quite complicated to have a background free training.

**Ethical Concerns:**

["NO or VERY MINOR ethics concerns only"]

**Final Justification:**

After checking the other reviews and the authors' rebuttal, I decided to increase my score. The authors addressed all my concerns and I think the paper is novel and interesting for the community.

**Limitations:**

Yes

**Quality:**

2

**Strengths And Weaknesses:**

Strengths
- Results. Policy adaptation happens very fast

Weaknesses
- Complex approach, a lot of hyper-parameters to deal with
- Requires a specific policy pretraining (black background)

---

> ### Author Rebuttal · Authors · 2025-07-31
>
> Thank you for providing valuable feedback. Below, we provide further clarification and results to address your concern and we hope these materials can enhance your evaluation of our paper.
>
> ## Q1: Too many hyper-parameters
> To address this concern, we highlight two key points from our experiments:
> - Stability: Our ablation studies (Section 5.3) demonstrate that FTR’s performance remains robust across hyper-parameters, suggesting low sensitivity to their exact values.
> - Consistency: The majority of experiments were conducted using the same fixed set of hyper-parameters (detailed in Table 3, Line 571). This also reduces the practical tuning burden.
>
> We acknowledge that the number of hyper-parameters might appear large at first glance, but their empirical stability and shared use across experiments mitigate the risk of overfitting or extensive tuning.
>
> ## Q2: Why is it necessary to have both a tracking and segmentation model given that they are used in an exclusive way? Is it related to cost trade-off for the usage of the VLM?
> We sincerely appreciate the reviewer's insightful question regarding our design. While the segmentation model alone could theoretically provide complete functionality, our empirical analysis in Section 4.2.2 (Line 176) and the ablation study on $T_{\text{sel}} = 1$ in Section 5.3 (Line 318) demonstrate that introducing the tracking model significantly improves segment selection consistency. This enhancement leads to more stable RL training.
>
> While the tracking mechanism does incidentally reduce VLM computation overhead (as noted by the reviewer), this gain is not our primary design motivation. Our key objective is to improve training stability through more consistent object selection across frames, as evidenced by our experimental results.
>
> ## Q3: Requires training the frozen policy using clean background. Would it be possible to remove such constraint?
> We appreciate the reviewer’s thoughtful question regarding the constraint of training the frozen policy in a clean-background setting. Indeed, this design choice is intentional and stems from our study’s core focus on domain adaptation under visual perturbations—specifically, investigating how a policy trained in a clean environment (source domain) can effectively adapt to environments with visual perturbation (target domain). Here are our justifications for this approach:
> - Many practical applications (e.g., robotics [1]) involve policies trained in structured/clean simulation settings that must later operate in noisy, real-world environments. Our setup mirrors this paradigm.
> - Many complex tasks, like robotic arm manipulation, are impractical or impossible to train directly in complex real-world environments as they require extensive interaction. While our method involves a two-stage training process, it offers much high sample efficiency. A low-level policy trained in a clean background can rapidly adapt to diverse real-world target environments. For instance, in the franka-reach task, the low-level policy requires 200k interactions with the simulation environment. Building on this, the high-level selector then only needs 50k real-world interactions to complete its training.
>
> While our current experiments focus on adaptation from clean to perturbed domains, as stated in the conclusion of the paper, the framework could be extended to other scenarios (e.g., training with synthetic perturbations). To demonstrate the feasibility of our approach under such conditions, we conducted experiments involving illumination perturbations. Due to time constraints, the experiments were performed with a single random seed.
>
> The table below presents our experimental results. We choose franka-reach as the environment. The performance of DrQ-v2 in a clean environment reported in our paper is 948.3, which represents the potential upper bound for FTR's performance.
>
> As shown in **row 1**, we initially evaluated the performance when the low-level policy was trained without illumination perturbations but deployed with their presence. By comparing this with **row 2**, which presents the adaptation performance of FTR in the target domain with both illumination and background perturbations, it is evident that FTR can effectively maintain the performance of the low-level policy under background disturbances, even when the low-level policy itself is not robust to illumination changes.
>
> Subsequently, we introduced varying directions of illumination perturbations during the training of the low-level policy in the source domain to enhance its robustness. As presented in **row 3**, the low-level policy's performance in environments with illumination perturbations significantly improved. **Row 4**, detailing FTR's adaptation performance in the target domain under both illumination and background perturbations, demonstrates that **FTR consistently maintained the performance of the low-level policy**. Furthermore, benefiting from the enhanced robustness of the low-level policy, the performance showed a notable improvement compared to the second row, closely approaching DrQ-v2's performance of 948.3 in a clean environment.
>
> | With background perturbation? | Low-level policy robust to illumination variation? | With FTR adaptation? | Performance |
> | --- | --- | --- | --- |
> | no | no | no | 611.5 |
> | yes | no | yes | 604.8 |
> | no | yes | no | 956.0 |
> | yes | yes | yes | 925.1 |
>
> We will emphasize these points in the paper’s limitations part to provide broader context. Thank you for raising this important discussion.
>
> ## Q4: How would you scale your approach to more complicated setting, e.g. navigation in Habit, robocasa, ManiSkill-HAB, Carla. In this case it seems quite complicated to have a background free training.
>
> First, we acknowledge that our experimental evaluation follows the same protocol established by prior works including the DMControl Generalization Benchmark [2], PAD [3], and SimGRL [4]. We sincerely appreciate the reviewer's insightful comment regarding the current limitations of the benchmark environments used in this line of research.
>
> To address the scalability challenge in more complex environments, taking navigation in Habit for example, we propose a multi-stage adaptation framework that maintains effectiveness while handling richer visual inputs:
> 1. Train a navigation policy in simulation using ground truth environmental information (e.g., full terrain maps)
> 2. Distill the terrain-aware policy into a vision-based policy using only egocentric RGB/RGB-D inputs
> 3. Apply our FTR approach to adapt to real-world environments, by automatically identifying and focusing on task-relevant visual inputs.
>
> ## References
>
> [1] Choi et al. On the Use of Simulation in Robotics: Opportunities, Challenges, and Suggestions for Moving Forward. Proceedings of the National Academy of Sciences. 2021.
>
> [2] Hansen et al. Generalization in Reinforcement Learning by Soft Data Augmentation. ICRA. 2021.
>
> [3] Hansen et al. Self-Supervised Policy Adaptation during Deployment. ICLR. 2021.
>
> [4] Song et al. A Simple Framework for Generalization in Visual RL under Dynamic Scene Perturbations. NeurIPS. 2024.

---

> ### Author Response · Authors · 2025-08-05
>
> Dear Reviewer,
>
> We truly appreciate the time and effort you have dedicated to reviewing our work. We hope the above clarifications and the additional experiments sufficiently addressed your concerns. Given the efforts we’ve made, I would be truly grateful if you could consider updating the score to reflect the newly added results and discussion. We would be happy to address any additional points you may have during the remaining time of the discussion phase.
>
> We thank you for engaging with us in the discussion.

---

> > ### Comment · Reviewer_R2gz · 2025-08-05
> >
> > Thank you for answering to my questions. I really appreciate the additional investigation you did in the rebuttal phase. I'm not against the paper to be accept and I may consider increasing the score based on the discussion with the other reviewers. However, for the time being I would prefer keeping the score as it is since Q3 and Q4 seem quite strong limitations to me.

---

### Official Review · Reviewer_WqUh · 2025-06-29

**Clarity:** 2
**Significance:** 2
**Originality:** 2
**Rating:** 5
**Confidence:** 3

**Summary:**

In this work the authors propose Focus-Then-Reuse (FTR) a framework for visual reinforcement learning under changes of background information. In particular, FTR is composed of two phases: (i) the focus stage, responsible for segmenting and tracking relevant information for the RL task (e.g., the agent, robot and objects on interest in the workspace) through pre-trained segmentation models and a trainable high-level segment selection policy; (ii) the reuse stage, where a standard RL pipeline is employed. The authors evaluate FTR in Mujoco environments and in three robotic tasks in simulation. The results show that their approach outperforms the baselines in robustness to visual disturbances in the background.

**Questions:**

I am willing to revise my score if the authors can address my concerns (see "Strengths and Weaknesses/Comments") and the following questions:

1) What is the performance of your method when dealing with standard visual perturbances (e.g., color changes, different lighting conditions and camera changes) and distractor objects?

2) Why did the authors not show the statistical results of the DrQ-v2 baseline? Also, in Table 1 which results are significantly better than the baselines?

**Ethical Concerns:**

["NO or VERY MINOR ethics concerns only"]

**Final Justification:**

The authors introduce a novel method to allow RL agents to execute policies under visual distractors, outperforming the previous baselines. My main concerns with the paper (limited novelty, limited applicability, lacking references to claims) were successfully addressed by the authors. As such, I have increased my score, and I believe the paper to be relevant to the RL community.

**Limitations:**

yes

**Quality:**

3

**Strengths And Weaknesses:**

**Strengths**:
- Paper is well-written and clear;
- Experimental results show a clear improvement over the selected baselines.

**Weaknesses**:
- Limited novelty (See "Comments" below);
- Some claims require support (See "Comments" below).

**Comments**:
- One major concern with the paper is the limited novelty of the work: using segmentation models to remove background distractors have been extensively explored in literature (e.g., [1, 2] for object-centric learning). From the discussion presented in the related work, and it is not clear where the novelty in the work lies. Moreover, the paper also introduces no novelty in terms of policy training as the authors employ standard PPO and DrQ-v2 algorithms for the focus and reuse stage policies.
- While not a weakness per se, I do have concerns regarding the significance of the work: the authors propose a method for removing visual background distractors by segmenting objects relevant to the task. This induces the policy to overfit to these elements and their visual appearance, unable to deal with potential color changes, different lighting conditions, different positions of the camera, amongst other visual perturbations that are common in real-world scenarios. It is also unclear how the method would perform in face of distractor objects in the scene (such as other cubes and/or robot manipulators, for example). While this is not the scope of the paper, the concern remains.
- Some claims in the paper are not supported by references in the current version of the paper: (i) in Lines 34-39, the authors claim that humans perform object-level filtering in decision making tasks, where object selection is initially driven by prior knowledge and subsequently fine-tuned by environmental feedback. It would improve the strength of the argument if the authors could support it with references. In line 225-226 the authors also claim that "the higher the accuracy of the segment selector, the greater the cumulative reward obtained by the low-level policy, and vice-versa". However, in the current version of the paper, the authors do not support this claim with any experimental validation.

**References**:

[1] - Shi, Junyao, et al. "Composing Pre-Trained Object-Centric Representations for Robotics From" What" and" Where" Foundation Models." 2024 IEEE International Conference on Robotics and Automation (ICRA). IEEE, 2024.

[2] - Wang, Ziyu, et al. "Generalizable visual reinforcement learning with segment anything model." arXiv preprint arXiv:2312.17116 (2023).

---

> ### Author Rebuttal · Authors · 2025-07-31
>
> We would like to thank you for your constructive feedback. We hope the following clarifications can further address your concerns and enhance your evaluation of our paper.
> ## Q1: Limited novelty
> - **Q1.1: Segmentation models have been explored in [1,2]**
>
> Thank you for pointing out these relevant works; we'll add citations to them in future version. We would like to highlight a key distinction between our approach and the method in [1]. It employs a rule-based strategy to select task-relevant segments. It lacks adaptability, since the rules cannot be dynamically adjusted to new environments. Compared with the method proposed in [1], our fusion of VLM supervision and RL finetuning for the segment selector is a novel approach, addressing the limitation of rule-based methods.
>
> The method in [2] requires post-training of SAM, which introduces additional overhead. Also, it relies on human annotation of task-relevant objects for SAM to generate segmentation masks. Our approach operates with SAM without finetuning and leverages SAM in a zero-shot manner.
>
> - **Q1.2: No novelty in policy training (PPO, DrQ-v2)**
>
> This paper aims to address the rapid adaptation of policies in environments with complex backgrounds. Instead of directly improving classic algorithms like PPO and DrQ-v2, we adopt a plug-in approach. We innovatively propose rapidly training a high-level selector during adaptation. Our method enables fast and efficient domain adaptation without needing to retrain the low-level policy $\pi_l$. Furthermore, it's compatible with existing visual RL algorithms, meaning any policy can serve as FTR's $\pi_l$ and be quickly deployed to the target domain.
>
> We would like to summarize the contributions of our paper:
> - We propose a novel policy adaptation framework employing an object selection mechanism to focus on task-relevant objects and then directly reusing a simulation-trained policy on them.
> - The focus stage and the reuse stage are highly decoupled, meaning that FTR is compatible with a range of generalization RL algorithms and able to handle complex distribution shift.
> - Our method integrates supervised learning and RL to synthesize VLM’s prior knowledge and environmental feedback, enabling efficient and effective adaptation.
>
> ## Q2: Policy may overfit to focused objects and backgrounds, unable to handle real-world perturbations like color changes, lighting, or camera position changes. What's the performance in such perturbations?
> To address this concern, we conduct additional experiments in a target domain featuring complex, real-world conditions. We choose franka-reach as the environment. Due to time constraints, the experiments were performed with one random seed.
>
> First, to verify FTR's ability to handle task-similar objects in the target domain that could easily be misidentified, we introduced cubes or uncontrolled robotic arms into the background and use FTR to adapt in this environment. The results, shown in the table below, indicate that even with such objects in the background, FTR can still filter out task-relevant objects using environmental reward, thereby maintaining high performance.
> |DrQ-v2 (clean background)|FTR (background perturbation)|FTR (background + another box)|FTR (background + another robot arm)|
> |-|-|-|-|
> |948.3|860.5|892.5|883.1|
>
> When deploying policies in other distractions like illumination, camera position, we propose two solutions: enhancing the robustness of $\pi_l$, and finetuning $\pi_l$ after adaptation.
>
> ### Enhancing the robustness of $\pi_l$
>
> As discussed in the conclusion section of our paper, improving the robustness of $\pi_l$ is a promising way to handle many other real-world disturbances. We introduce illumination perturbation during low-level policy $\pi_l$ training to enhance robustness.
>
> **Row 1** shows performance without illumination-robust training, while **row 2** demonstrates FTR's adaptation under both illumination and background perturbations, maintaining the performance of $\pi_l$ despite illumination sensitivity.
>
> Subsequently, we introduce varying directions of illumination perturbations during the training of $\pi_l$ in the source domain to enhance its robustness. As presented in **row 3**, $\pi_l$'s performance improved significantly. **Row 4** shows FTR's adaptation performance under both illumination and background perturbations, demonstrates that **FTR consistently maintained the performance of $\pi_l$**. Furthermore, benefiting from the enhanced robustness of $\pi_l$, the performance showed a notable improvement compared to row 2, closely approaching DrQ-v2's performance of 948.3 (as a potential performance upper bound for FTR performance) in the source domain.
> |With background perturbation?|$\pi_l$ robust to illumination variation?|With FTR adaptation?|Performance|
> |-|-|-|-|
> |no|no|no|611.5|
> |yes|no|yes|604.8|
> |no|yes|no|956.0|
> |yes|yes|yes|925.1|
>
> ### Finetuning $\pi_l$ after adapting
>
> When $\pi_l$ lacks robustness or is deployed in an unforeseen perturbed environment, we show **finetuning $\pi_l$ after adapting** (while fixing the selector) can further improve performance.
>
> To demonstrate this, we selected two types of target domain perturbations:
> 1. **Camera rotation**: 15° around the z-axis.
> 2. **Base inclination angle**: The robot arm's base is tilted 15° horizontally.
>
> $\pi_l$ was trained in an unperturbed source domain, rendering it non-robust to these specific target domain perturbations.
>
> After **training the selector for 200k steps**, we **fix its parameters and unfreeze $\pi_l$'s parameters** to **finetune $\pi_l$ for 50k steps** using environmental reward. As the table below shows, the adapted FTR performance initially falls short of the potential upper bound of 948.3. Nevertheless, after the 50k-step finetuning, the performance recovers to a commendable level.
> ||After FTR adaptation|After finetuing $\pi_l$|
> |-|-|-|
> |Camera angle|755.1|903.3|
> |Base inclination angle|776.0|896.5|
>
> We posit that the reason for the effectiveness of the "adapt + finetune" process and the performance maintenance with non-robust $\pi_l$ is same: **The selector's RL training does not impose stringent requirements on the optimality of $\pi_l$**. The selector can be guided towards correct learning as long as $\pi_l$ satisfies a key condition: the reward for a correctly chosen action by the selector is greater than that for an incorrectly chosen one. As shown in the experiment of Q3, this condition can be easily satisfied. Once the selector learns the correct selection patterns, finetuning $\pi_l$ based on the filtered images becomes highly efficient.
>
> ## Q3: Claims not supported by references
>
> - **Lines 34-39: "humans perform object-level filtering in decision making tasks, where object selection is initially driven by prior knowledge and subsequently finetuned by environmental feedback"**
>
>     We sincerely appreciate your constructive comments.We would like to cite the following literature that supports our claims above:
>     1. Reference [3] draws on research on the neurobiology of attention and suggests that human cognition involves selective attention mechanisms, where only the most important signals are processed for action.
>     2. Reference [4] offers theoretical support through its summarization on research on the roles of prior knowledge in learning, noting that "Learning proceeds primarily from prior knowledge" and "Prior knowledge is the bane of learning. Mere absorption cannot account for the revolutionary changes".
>
>     We will add these citations in the future version of the paper.
> - **Lines 225-226: "the higher the accuracy of the segment selector, the greater the cumulative reward obtained by the low-level policy, and vice-versa"**
>
>     We conduct supplementary experiments in franka-reach environment to validate the relationship between segment selection accuracy and cumulative reward in an episode over five random seeds. The results in the table below demonstrate a clear correlation: When the segment selector accurately focuses on task-relevant objects, $\pi_l$ achieves the highest rewards. Conversely, incorrect selection leads to substantially lower rewards.
>     We will discuss these experimental results and their implications in the appendix.
>     |Only task-relevant objects|Task-relevant objects + 1 irrelevant object|Task-relevant objects + 2 irrelevant objects|All objects|All task-irrelevant objects|
>     |-|-|-|-|-|
>     |**892.4±22.9**|570.9±223.1|495.3±259.3|234.4±282.9|9.6±17.2|
>
> ## Q4: Where are the statistical results of DrQ-v2? & Which results are significantly better than the baselines in Table 1?
> We appreciate the reviewer's attention to methodological details. To clarify our experimental design for DrQ-v2, as detailed in Section 5.1, we conduct three independent runs in the clean environment using DrQ-v2, selecting the best-perfroming policy as $\pi_l$ in FTR. This represents the potential upper bound for domain adaptation performance.
>
> The performance gap between DrQ-v2 and other baselines in Table 1 stems from their different evaluation conditions: while DrQ-v2 was evaluated in the clean source domain, all other methods were tested in visually perturbed target environments. We recognize that this methodological distinction could have been more prominently highlighted, and will add an explanatory footnote to Table 1 in the revised version.
>
> ## References
> [1] Shi et al. Composing Pre-Trained Object-Centric Representations for Robotics From What and Where Foundation Models. ICRA. 2024.
>
> [2] Wang et al. Generalizable Visual Reinforcement Learning with Segment Anything Model. arXiv. 2023.
>
> [3] Stevens et al. The Role of Selective Attention on Academic Foundations: A Cognitive Neuroscience Perspective. Developmental Cognitive Neuroscience. 2012.
>
> [4] Roschelle J. Learning in Interactive Environments: Prior Knowledge and New Experience. Exploratorium Institute for Inquiry. 1997.

---

> > ### Comment · Reviewer_WqUh · 2025-08-01
> > **Acknowledgment of Author's Rebuttal**
> >
> > Dear authors,
> >
> > Thank you for the rebuttal and the effort conducting the extra experiments. Since you clarified my points and strengthen the paper with the additional experiments, I will raise my score to 5.
> >
> > As a final comment, I would incentivize the authors to run more statistics (i.e., more seeded-runs) on the additional experiments, as one single-seed provides limited confidence in the results. Additionally, always try to provide statistical significance results when conducting RL experiments (see [1])., for example providing 95% CI.
> >
> > [1] - Patterson, Andrew, et al. "Empirical design in reinforcement learning." Journal of Machine Learning Research 25.318 (2024): 1-63.

---

> > > ### Author Response · Authors · 2025-08-01
> > >
> > > Dear Reviewer,
> > >
> > > Thank you very much for raising your score and for your constructive comments.
> > >
> > > Following your advice, we will conduct additional experiments with more random seeds to strengthen the statistical reliability of our results. We aim to complete as many of these experiments as possible within this week and will submit the updated results promptly. We will also include them in the future version of the paper.
> > >
> > > Thank you again for your time and for your insightful feedback.

---

> > > ### Author Response · Authors · 2025-08-05
> > >
> > > We have conducted additional experiments with 3 different random seeds. The tables below present the results of the average performance and the standard deviation of the final checkpoint. The updated results consistently support our original findings.
> > >
> > > |FTR (background + another box)|FTR (background + another robot arm)|
> > > |-|-|
> > > |913.0±18.5|871.8±38.6|
> > >
> > > |With background perturbation?|$\pi_l$ robust to illumination variation?|With FTR adaptation?|Performance|
> > > |-|-|-|-|
> > > |no|no|no|659.3±52.5|
> > > |yes|no|yes|599.1±62.1|
> > > |no|yes|no|952.3±4.1|
> > > |yes|yes|yes|929.5±11.6|
> > >
> > > ||After FTR adaptation|After finetuing $\pi_l$|
> > > |-|-|-|
> > > |Camera angle|733.3±15.7|904.6±8.7|
> > > |Base inclination angle|716.2±51.0|906.0±13.7|

---

### Official Review · Reviewer_8K8e · 2025-06-30

**Clarity:** 3
**Significance:** 3
**Originality:** 3
**Rating:** 5
**Confidence:** 4

**Summary:**

The paper introduces a framework called Focus-Then-Reuse (FTR) to address the challenge of deploying reinforcement learning policies trained in simulation to real-world environments with visual perturbations. FTR has two stages: (1) focus stage: dynamically selects task-relevant objects using a trainable segment selector (trained via RL + VLM supervision), (2) reuse stage: executes actions with a fixed, pre-trained policy on the filtered observations. FTR outperforms baselines like SimGRL (generalisation) and PAD (adaptation), especially in complex visual perturbations.

**Questions:**

1. How does FTR handle cases where SAM2 or the tracker fails to segment/task-relevant objects? Since the low-level policy depends on the selected segments, what is the impact on performance in real-world settings where such failures are common?

2. Are the initial segmentation masks (or tracking updates) generated by SAM2 without prompts? Could the target object (e.g., a robot arm or manipuland) be oversegmented or contaminated with background pixels? If so, how does this affect policy performance?

3. Instead of feeding raw pixels to the low-level policy, have you explored using SAM2’s object features (e.g., mask embeddings) as inputs? Would this introduce bottlenecks (e.g., latency, feature misalignment) compared to the current pipeline?

**Ethical Concerns:**

["NO or VERY MINOR ethics concerns only"]

**Final Justification:**

See the rebuttal comment.

**Limitations:**

Yes

**Paper Formatting Concerns:**

No major formatting issues.

**Quality:**

3

**Strengths And Weaknesses:**

**Strengths:**

1. Integration of foundation models into RL:
    - The paper innovatively leverages large-scale pretrained models (Qwen-VL and SAM 2) to address visual RL challenges, bridging the gap between high-level perception and low-level control. This is a promising direction, as foundation models offer rich prior knowledge but remain underutilised in RL for dynamic decision-making.
    - The implementation (released in the supp.) is practically valuable, providing a valuable pipeline for combining VLMs with RL. This could inspire follow-up work on foundation-model-augmented RL.

2. The fusion of VLM supervision and RL fine-tuning for the segment selector is a novel approach. The VLM provides an initial object-selection prior, while RL adapts it to task-specific rewards. The dynamic weighting mechanism (transitioning from SL to RL) is well-justified and could generalise to other hierarchical RL settings.

**Weakness:**

1. Sim-to-real gaps in the high-level policy:
    - While the focus stage mitigates visual noise, the high-level policy (segment selector) is still trained purely in simulation. Real-world perceptual challenges (e.g., lighting changes, motion blur) may degrade its performance, necessitating additional adaptation.
    - The paper does not test FTR on real-world deployment or quantify how the segment selector’s sim-trained priors (from Qwen-VL/SAM) transfer to real images. This limits claims about real-world readiness.

2. Dependence on a pretrained low-level policy:
    - FTR assumes access to a low-level policy trained in a clean simulation, which may not exist for many real-world systems. Even with a "perfect" simulation twin, the filtered observations may still differ from the simulation’s training distribution.
    - The low-level policy is frozen during adaptation; FTR cannot correct for dynamics mismatches (e.g., sim-to-real mechanical discrepancies).

---

> ### Author Rebuttal · Authors · 2025-07-31
>
> Thank you for your constructive comments and suggestions. Below are further clarifications for your concerns.
> ## Q1: Sim-to-real gaps in the high-level policy
> - **Q1.1: Simulation-trained high-level policy may degrade in real-world perception.**
>
> We would like to clarify that our experiment follows the study protocol of DMControl Generalization Benchmark [1], PAD [2], and SimGRL [3]. To address your concern, we conduct further experiments in a target domain with more intricate, realistic conditions. We choose franka-reach as the environment. Due to time constraints, these experiments were performed with one random seed.
>
> First, to verify FTR's ability to handle task-similar objects in the target domain that could easily be misidentified, we introduced cubes or uncontrolled robotic arms into the background and use FTR to adapt in this environment. The results, shown in the table below, indicate that even with such objects in the background, FTR can still filter out task-relevant objects using environmental reward, thereby maintaining high performance.
> |DrQ-v2 (clean background)|FTR (background perturbation)|FTR (background + another box)|FTR (background + another robot arm)|
> |-|-|-|-|
> |948.3|860.5|892.5|883.1|
>
> Going a step further, we evaluated FTR's illumination generalization. To enhance the adapted policy's performance, as discussed in the conclusion of our paper, we introduce illumination perturbation during low-level policy $\pi_l$ training to enhance robustness.
>
> **Row 1** shows performance without illumination-robust training, while **row 2** demonstrates FTR's adaptation under both illumination and background perturbations, maintaining the performance of $\pi_l$ despite illumination sensitivity.
>
> Subsequently, we introduced varying directions of illumination perturbations during the training of $\pi_l$ in the source domain to enhance its robustness. As presented in **row 3**, $\pi_l$'s performance improved significantly. **Row 4** shows FTR's adaptation performance under both illumination and background perturbations, demonstrates that **FTR consistently maintained the performance of the low-level policy $\pi_l$**. Furthermore, benefiting from the enhanced robustness of $\pi_l$, the performance showed a notable improvement compared to row 2, closely approaching DrQ-v2's performance of 948.3 (as a potential performance upper bound for FTR performance) in the source domain.
> |With background perturbation?|$\pi_l$ robust to illumination variation?|With FTR adaptation?|Performance|
> |-|-|-|-|
> |no|no|no|611.5|
> |yes|no|yes|604.8|
> |no|yes|no|956.0|
> |yes|yes|yes|925.1|
>
> - **Q1.2: The paper does not test FTR on real-world deployment or quantify how the segment selector’s priors from VLM/SAM transfer to real images**
>
> Due to experimental constraints, testing FTR in real-world scenarios is beyond our current scope. As an alternative validation, we perform the following evaluations on six real robotic arm videos from Open-X-Embodiment [4] to assess:
> - **SAM2's segmentation robustness:** SAM2 successfully segments the robotic arm in almost all cases, demonstrating strong real-world applicability. Minor limitations are observed in gripper detection with a failure rate of 4.3%, which might sightly affect the performance of FTR in real-world.
> - **VLM’s recognition reliability with segments:** The VLM demonstrates reliable identification of robotic arms in real-world scenarios, achieving an accuracy of 90.9%. This performance aligns with our experimental result in appendix B.2, confirming the VLM's capability in real-world scenarios.
>
> These demonstrate promising real-world transfer potential, though real-world deployment testing would further strengthen our claims.
> ## Q2: Dependence on a pretrained low-level policy $\pi_l$
> - **Q2.1: FTR assumes $\pi_l$ trained in clean simulation, which may not exist for real-world systems.**
>
> Here are our justifications for this approach:
> - Many practical applications (e.g., robotics [5]) involve policies trained in structured/clean simulation settings that must later operate in noisy, real-world environments. Our setup mirrors this paradigm.
> - Many complex tasks, like robotic arm manipulation, are impractical or impossible to train directly in complex real-world environments as they require extensive interaction. While our method involves a two-stage training process, it offers much high sample efficiency. A $\pi_l$ trained in a clean background can rapidly adapt to diverse real-world target environments. For instance, in the franka-reach task, $\pi_l$ requires 200k interactions with the simulation environment. Building on this, the high-level selector then only needs 50k real-world interactions to complete its training.
>
> While our current experiments focus on adaptation from clean to perturbed domains, as stated in the conclusion of the paper, the framework could be extended to other scenarios (e.g., training with synthetic perturbations). To demonstrate the feasibility of our approach under such conditions, we conducted experiments involving illumination perturbations, as shown in Q1.
>
> - **Q2.2: Frozen $\pi_l$ during adaptation limits dynamics mismatch correction**
>
> We acknowledge that while Q1's source-domain perturbation enhance $\pi_l$'s robustness, performance may degrade if perturbations don't fully cover target domain variations. However, we show **finetuning $\pi_l$ after adapting** (while fixing the selector) can further improve performance.
>
> To demonstrate this, we selected two types of target domain perturbations:
> 1. **Camera rotation**: 15° around the z-axis.
> 2. **Base inclination angle**: The robot arm's base is tilted 15° horizontally.
>
> $\pi_l$ was trained in an unperturbed source domain, rendering it non-robust to these specific target domain perturbations.
>
> After **training the selector for 200k steps**, we **fix its parameters and unfreeze $\pi_l$'s parameters** to **finetune $\pi_l$ for 50k steps** using environmental reward. Due to time constraints, these experiments were performed with one random seed. As the table below shows, the adapted FTR performance initially exhibits a gap compared to the potential upper bound of 948.3. Nevertheless, after the 50k-step finetuning, the performance recovers to a commendable level.
>
> We posit that this **"adapt + finetune" process is effective because the selector's RL training does not impose stringent requirements on the optimality of $\pi_l$**. The selector can be guided towards correct learning as long as $\pi_l$ satisfies a key condition: the reward for a correctly chosen action by the selector is greater than that for an incorrectly chosen one. This condition is typically met even in the presence of a domain gap. Once the selector learns the correct selection patterns, finetuning $\pi_l$ based on the filtered images becomes highly efficient.
> ||After FTR adaptation|After finetuing $\pi_l$|
> |-|-|-|
> |Camera angle|755.1|903.3|
> |Base inclination angle|776.0|896.5|
>
> ## Q3: How does FTR handle SAM2's failures? What's the performance impact in real-world?
> We appreciate this important question regarding failure cases in our pipeline. In our experiments, SAM2 demonstrated near-perfect segmentation accuracy on the benchmark, which is why we didn't emphasize this in the paper. Also, the evaluations on real-world videos from the Open-X-Embodiment dataset stated in Q1 shows that SAM2 remains robust in real-world scenarios.
>
> We acknowledge that the performance of FTR depends on SAM2, as the pipeline depends on proper object segmentation in the focus stage. Should SAM2 fail to produce valid segmentation masks (a rare occurrence in our experiments but theoretically possible), FTR would consequently be unable to function as intended. We acknowledge this could have been more clearly stated as part of our system's assumptions in the Limitations part.
>
> ## Q4: Prompt requirements in SAM2.
> In our paper, SAM2 generates segmentation masks and updates tracking without prompt.
>
> ## Q5: Could the target object be oversegmented or contaminated with background pixels? If so, how does this affect policy performance?
> Our method filters the masks using the method in [6]: discard smaller segments if they overlap more than 95% with a larger segment, and select up to nine segments in descending order of size. In this case, the target objects are barely oversegmented. As shown in the visualization of franka-reach in supplementary, the target object do sometimes be contaminated with background pixels. The experimental results on franka-reach in Section 5.2 indicate that this hardly affects performance.
>
> ## Q6: What about using SAM2’s object features as inputs? Would this introduce bottlenecks (e.g., latency, feature misalignment) compared to the current pipeline?
> For $\pi_l$, the task of decision making based on filtered images isn't particularly difficult. Given that DrQ-v2 already achieves good performance, incorporating SAM2's object features as input isn't necessary.
>
> Furthermore, because SAM2's training objective is object segmentation and tracking, it doesn't focus on other information like color or relative position. Therefore, using SAM2's object features as input might cause $\pi_l$ to lose crucial information, leading to a performance drop.
> ## References
> [1] Hansen et al. Generalization in Reinforcement Learning by Soft Data Augmentation. ICRA. 2021.
>
> [2] Hansen et al. Self-Supervised Policy Adaptation during Deployment. ICLR. 2021.
>
> [3] Song et al. A Simple Framework for Generalization in Visual RL under Dynamic Scene Perturbations. NeurIPS. 2024.
>
> [4] O’Neill et al. Open X-Embodiment: Robotic Learning Datasets and RT-X Models: Open X-Embodiment Collaboration. ICRA. 2024.
>
> [5] Choi et al. On the Use of Simulation in Robotics: Opportunities, Challenges, and Suggestions for Moving Forward. Proceedings of the National Academy of Sciences. 2021.
>
> [6] Chen et al. Focus-Then-Decide: Segmentation-Assisted Reinforcement Learning. AAAI. 2024.

---

> > ### Comment · Reviewer_8K8e · 2025-08-06
> >
> > I appreciate the authors' efforts in the rebuttal. Most of my concerns are explained, and this work does provide a practical way to perform automatic segmentation selection, which I believe is quite useful for many purposes. I will raise my score to 5.

---

> > > ### Author Response · Authors · 2025-08-06
> > >
> > > Dear Reviewer,
> > >
> > > Thank you very much for your time, for your insightful feedback, and for raising your score.

---

### Official Review · Reviewer_HyXv · 2025-07-02

**Clarity:** 4
**Significance:** 3
**Originality:** 2
**Rating:** 5
**Confidence:** 4

**Summary:**

The authors propose a VLM/segmentation-model assisted background segmentation scheme to help combat visual distractions in reinforcement learning. First, an optimal control policy is trained in the clean simulation environment. Then, an adaptation phase is conducted in the target environment. SAM-2 is used to propose candidate image segments to be provided to the frozen simulation control policy. To choose the image segments, a selection policy is trained in the target environments via downstream target environment RL rewards (collected by the control policy). To jumpstart training, the selection policy is additionally trained with supervised learning to match VLM suggestions in early stages of training. Segment selection is only performed occasionally, and currently selected segments are tracked between frames using the pretrained segmentation model. This method outperforms a variety of adaptation, generalization, and target-environment training approaches.

**Questions:**

It would be helpful to add a sanity check for all baselines (not just DrQ-v2) on the clean environment to show that they all can correctly learn an optimal policy and that any failures are due to the distractions. Could this be added?

**Ethical Concerns:**

["NO or VERY MINOR ethics concerns only"]

**Final Justification:**

All of my concerns originally mentioned in this review have been adequately addressed. I recommend this paper for acceptance.

**Limitations:**

The authors state that other methods, when facing unseen disturbances, may drop performance significantly.
They should add more clear acknowledgements that their method still relies on the simulation policy generalizing to the target environment appearance for the segmented foreground objects. Likewise, this method can also drop in performance if the in-focus objects have a different appearance (lighting, texture, occlusion, etc.) compared to the simulation.

Another downside is that they require deployment-time segmentation/tracking, which can limit control frequency. Other baselines as well as [1] do not require this, and this should be mentioned.

**Paper Formatting Concerns:**

No formatting concerns.

**Quality:**

3

**Strengths And Weaknesses:**

The paper is well-written, proposing a straightforward and effective idea that other practitioners should be able to readily reproduce.
My only notable complaint is that the novelty is somewhat incremental due to VLMs having been used for background segmentation before, for example in [1], however the introduction of the segment-selection policy trained in the target-domain is new and significant, so I'm leaning towards acceptance.

Experiments and ablations seem sufficient, except for one concern listed in *Questions*.

This is a missing reference that also uses VLMs to suggest segmentation masks while training with domain-randomized video backgrounds:

[1] Kim et al. Make the Pertinent Salient: Task-Relevant Reconstruction for Visual Control with Distractions. 2024.

---

> ### Author Rebuttal · Authors · 2025-07-31
>
> We sincerely appreciate your insightful feedback, which has greatly improved our work. We hope the following clarifications can further address your concerns.
>
> ## Q1: Sanity check for all baselines
>
> The table below shows the results of the baselines on the clean environment. Due to time constraints, the experiments were performed with a single random seed. PAD and SimGRL perform well on almost all the tasks, while Q$^2$-learning performs decently except fp and fd. We will include the sanity check in the appendix.
> | Task | PAD | Q$^2$-learning | SimGRL |
> | --- | --- | --- | --- |
> | ps | 964 | 944.0 | 904 |
> | cs | 845 | 874.9 | 873 |
> | fs | 927 | 390.3 | 986 |
> | hs | 789.1 | 863.2 | 872.0 |
> | hh | 116.2 | 184.7 | 145.7 |
> | cr | 380 | 372.2 | 322 |
> | ww | 895 | 660.7 | 923 |
> | wr | 311.0 | 288.5 | 338.1 |
> | fr | 939.4 | 928.8 | 955.1 |
> | fp | 100.7 | 60.4 | 84.1 |
> | fd | 148.0 | 20.1 | 162.4 |
>
> ## Q2: The novelty is incremental for background segmentation[1]
>
> We agree that segmentation model have indeed been explored for background segmentation in prior works, including [1]. The method in [1] primarily focuses on using segmentation masks in the auxiliary task to improve representation learning in distracting environments. Compared with the method proposed in [1], our **fusion of VLM supervision and RL finetuning in the target domain for the segment selector is a novel approach**.
>
> We would like to summarize the contributions of our paper:
> - We propose a novel policy adaptation framework employing an object selection mechanism to focus on task-relevant objects and then directly reusing a simulation-trained policy on them.
> - The focus stage and the reuse stage are highly decoupled, meaning that FTR is compatible with a range of generalization RL algorithms and able to handle complex distribution shift.
> - Our method integrates supervised learning and RL to synthesize VLM’s prior knowledge and environmental feedback, enabling efficient and effective adaptation.
>
> We are grateful to you for your careful reading of our paper and for pointing out the missing reference on VLM and segmentation masks. We will include a dedicated discussion of this aspect in the Related Work section of our future version of the paper.
>
> ## Q3: Needs clearer acknowledgment that the method relies on simulation policy generalizing to target environment foreground appearance. Performance may drop if in-focus objects' appearance (lighting, texture, occlusion, etc.) differs from simulation.
>
> We appreciate the reviewer for pointing out this limitation. We acknowledge that when the appearance of in-focus objects in the target domain differs from that in the source domain, the adaptation performance of FTR relies on the generalization capability of the low-level policy $\pi_l$. To demonstrate the feasibility of our approach under such conditions, we conducted experiments involving illumination perturbations. Due to time constraints, the experiments were performed with a single random seed.
>
> The table below presents our experimental results. We choose franka-reach as the environment. The performance of DrQ-v2 in a clean environment reported in our paper is 948.3, which represents the potential upper bound for FTR's performance.
>
> As shown in **row 1**, we initially evaluated the performance when $\pi_l$ was trained without illumination perturbations but deployed with their presence. By comparing this with **row 2**, which presents the adaptation performance of FTR in the target domain with both illumination and background perturbations, it is evident that FTR can effectively maintain the performance of $\pi_l$ under background disturbances, even when $\pi_l$ itself is not robust to illumination changes.
>
> Subsequently, we introduced varying directions of illumination perturbations during the training of $\pi_l$ in the source domain to enhance its robustness. As presented in **row 3**, $\pi_l$'s performance in environments with illumination perturbations significantly improved. **Row 4**, detailing FTR's adaptation performance in the target domain under both illumination and background perturbations, demonstrates that **FTR consistently maintained the performance of the low-level policy $\pi_l$**. Furthermore, benefiting from the enhanced robustness of $\pi_l$, the performance showed a notable improvement compared to the second row, closely approaching DrQ-v2's performance of 948.3 in a clean environment.
>
> | With background perturbation? | Low-level policy robust to illumination variation? | With FTR adaptation? | Performance |
> | --- | --- | --- | --- |
> | no | no | no | 611.5 |
> | yes | no | yes | 604.8 |
> | no | yes | no | 956.0 |
> | yes | yes | yes | 925.1 |
>
> ## Q4: Require deployment-time segmentation/tracking
>
> We acknowledge the reviewer's concern regarding the deployment-time overhead introduced by segmentation/tracking. While this indeed incurs an additional ~50% computational cost during evaluation as we measured, we argue that the trade-off is justified due to significant improvements. Specifically, our approach demonstrates superior performance compared to the baseline, as shown in Section 5.2. If long-term policy deployment in noisy environments is required, one way to reduce computational overhead is to collect interaction data and distill a policy that relies solely on noisy inputs. We acknowledge this could have been more clearly stated as part of our system's assumptions in the Limitations part.
>
> ## References
>
> [1] Kim et al. Make the Pertinent Salient: Task-Relevant Reconstruction for Visual Control with Distractions. arXiv. 2024.

---

> > ### Comment · Reviewer_HyXv · 2025-08-08
> >
> > I thank the authors for their rebuttal. All of my concerns have been adequately addressed. I recommend this paper for acceptance.

---

### Decision · Program_Chairs · 2025-09-17

**Decision:**

Accept (poster)

**Comment:**

This paper proposes a two-stage framework to help reinforcement learning agents adapt from a clean simulation to environments with distractions, using a trainable object selector to focus on task-relevant objects, followed by pre-trained policy on filtered input.
The paper's main strength is its practical integration of large pre-trained models (a VLM and a segmentation model) into a reinforcement learning adapation pipeline. A primary weakness is the reliance on a low-level policy trained in a clean environment without distractions. While the authors show this can be made more robust with domain randomization or a final finetuning step, it remains a core assumption of the approach.

The discussion period was productive, reviewers raised important questions about the method's novelty (HyXv, WqUh) and ability to handle perturbations beyond background noise (8K8e, WqUh). The authors provide a comprehensive rebuttal including new results demonstrating robustness to distractor objects and lighting variations as well as short finetuning stage to address dynamics mismatches. The response addressed the reviewers' main concerns leading to positive consensus.